

**Evaluating landslide response in seismic and rainfall regime:**
**A case study from the SE Carpathians, Romania**
[1]Vipin Kumar, [1]Léna Cauchie, [1]Anne-Sophie Mreyen, [2]Mihai Micu, [1]Hans-Balder Havenith
[1]Georisk and Environment, Department of Geology, University of Liege, Belgium
[2]Institute of Geography, Romanian Academy, Bucharest, Romania
Correspondence: Vipin Kumar (v.chauhan777@gmail.com)
**Abstract**
There have been many studies exploring the rainfall induced slope failures in the earthquake
affected terrain. However, studies evaluating the potential effects of both landslide triggering
factors; rainfall and earthquake have been infrequent despite the rising global landslide
mortality risk. The SE Carpathians, which have been subjected to many large historical
earthquakes and changing climate and thus resulting in frequent landslides, is one such region
that is least explored in this context. Therefore, a massive (~9.1 Mm²) landslide, situated
along the Basca Rozilei River, in the Vrancea Seismic Zone, SE Carpathians is chosen as a
case study area to achieve the aforesaid objective. The present state of slope reveals the Factor
of Safety in a range of 1.17-1.32 with a static condition displacement of 0.4-4 m that reaches
up to 8-60 m under dynamic (earthquake) condition. The Groundwater (GW) effect further
decreases the Factor of Safety and increases the displacement. Ground motion amplification
enhances the possibility of slope surface deformation and displacements. The debris flow
prediction, implying the excessive rainfall effect, reveals a flow having 9.0-26.0 m height and
2.1-3.0 m/sec velocity along the river channel. The predicted extent of potential debris flow is
found to follow the trails possibly created by previous debris flow and/or slide events.
**Key words**: Landslide; Earthquake; Rainfall; Slope Stability; Runout; SE Carpathians.



## 1 Introduction

Landslides, though a normal process of hillslope erosion, pose socio-economic risk to human life and infrastructure (Froude and Petley 2018; Pollock and Wartman 2020; Kumar et al. 2021). Despite the rising global landslide mortality risk, effective evaluation of disastrous influences of landslides has been infrequent (Sassa 2015; Haque et al. 2019; Klimes et al. 2019). Such evaluation approaches could be regional (susceptibility/hazard/risk/vulnerability) or local (slope stability, runout prediction, monitoring/change-detection mapping) (Fell and Hartford 1997; Westen et al. 2006; Margottini et al. 2013; Hungr 2018). However, effectiveness in such approaches cannot be justified until the main landslide triggering factors; rainfall and earthquake are evaluated together. Despite the numerous case studies of rainfall induced slope failures in the earthquake affected terrain (Lin et al. 2006; Helmstetter et al. 2010; Tang et al. 2011; Durand et al. 2018; Bontemps et al. 2020), studies predicting the potential effects of both factors have been relatively rare. Necessity of such studies becomes more critical in view of an annual average of >4000 landslide related deaths worldwide in the last decade (Froude and Petley 2018).

Owing to the capability to represent the progressive deformation in the slope under various loading conditions, numerical modeling based analysis can be considered as one of the few approaches for effective evaluation of slope instability and associated risk (Jing 2003; Fenton and Griffiths 2008). Though the continuum modelling based approaches have been common for local scale evaluation of hillslope response (Griffiths and Lane 1999; Jamir et al. 2017; Kumar et al. 2018; 2021), their limitations in estimating large strain, particularly during the dynamic analysis makes the discontinuum modeling better option (Havenith et al.2003; Bhasin and Kaynia 2004). Apart from the stability evaluation, prediction of potential run-out during the slope failure constitutes a principal risk evaluation approach (Hungr et al. 1984; Hutter et al. 1994; Rickenmann and Scheidl 2013). Among different types of landslides, debris flows have shown the maximum outreach, relatively more fatality, and secondary effects like river damming and subsequent outburst flood (Jakob et al. 2005; Ding et al. 2020; Kumar et al. 2021). Among different run-out prediction approaches, dynamic model based Rapid Mass Movement Simulation (RAMMS) (Christen et al. 2010), Flo-2D (O'Brien et al. 1993), and MassMov2D (Beguer´ıa et al. 2009) have been relatively more useful (Rickenmann and Scheidl, 2013; Kumar et al. 2021).



In view of these understandings, the present study aimed to infer the potential response of a landslide slope under the seismic and extreme rainfall conditions using stability evaluation and runout simulation. Such simulations/modeling outputs depend upon certain input parameters and criteria, the values of which might be affected by uncertainties due to nonlinear behavior of material. Therefore, a parametric analysis is also performed to evaluate the uncertainty. In order to achieve the aforementioned objectives, a massive (~9.1 Mm²) landslide in the Vrancea Seismic Zone, SE Carpathians is chosen as a case study area. The region has been subjected to frequent earthquakes and relatively wet climatic conditions that induce frequent landslides and related socio-economic losses (Micu et al. 2013; 2016; Micu, 2019; Mreyen et al. 2021).

## 2 Study area

### 2.1 Geological setting & geomorphology

The landslide is situated at latitude 45° 30' 23" N, longitude 26° 25' 05" E along the Basca Rozilei River in the SE Carpathians, Romania (Fig. 1). The slope is composed of shale belonging to the Miocene thrust belt that separates the external foredeep in the north, east, and south-east from the inner Carpathians mountain ranges. Thrust faults, strike-slip faults, and folds traverse the region in and around the vicinity of landslide slope. The origin of these structural features has been related to the Eocene-Miocene collision of Alcapa and Tisza-Dacia plates against the Bohemian and Moesian promontories that gave rise to the Carpathians Mountain (Tischler et al. 2008). The SE part of the Carpathians, however, is still uplifting at a rate of 3-8 mm/yr. due to the foreland coupling of the converging plates (Pospisil and Hipmanova 2012; Mațenco 2017).

The landslide toe along the river hosts the 'Varlaam' village (Fig. 1, 2a). The landslide has a slope gradient ranging between 15°-20° and encompasses an area of ~9.1 Mm². The landslide-affected area is covered by shrubs and scattered trees towards its flanks and with grasslands in the inner parts, mainly used as pastures and hayfields. The landslide crown region has a depression that might be a surficial imprint of the paleo-detachment (or depletion zone) (Fig. 2b). Near the right (or southern) flank, a seasonal flow channel (or gully) emerges near the paleo-detachment depression and finally merges at the river channel (Fig. 2c). Near the left (or northern) flank, slope surface comprises flow relics, possibly of paleo-debris flow and/or slide events (Fig. 2d), as also inferred from loose/unconsolidated deposit at the slope toe (Fig.


2e). This flow deposit is noted to develop 100-150 m wide minor scarps (Fig. 2e). Such scarps
may further grow and result in the debris flows during extreme rainfall and/or earthquake
events and hence pose a risk to the nearby human settlement.

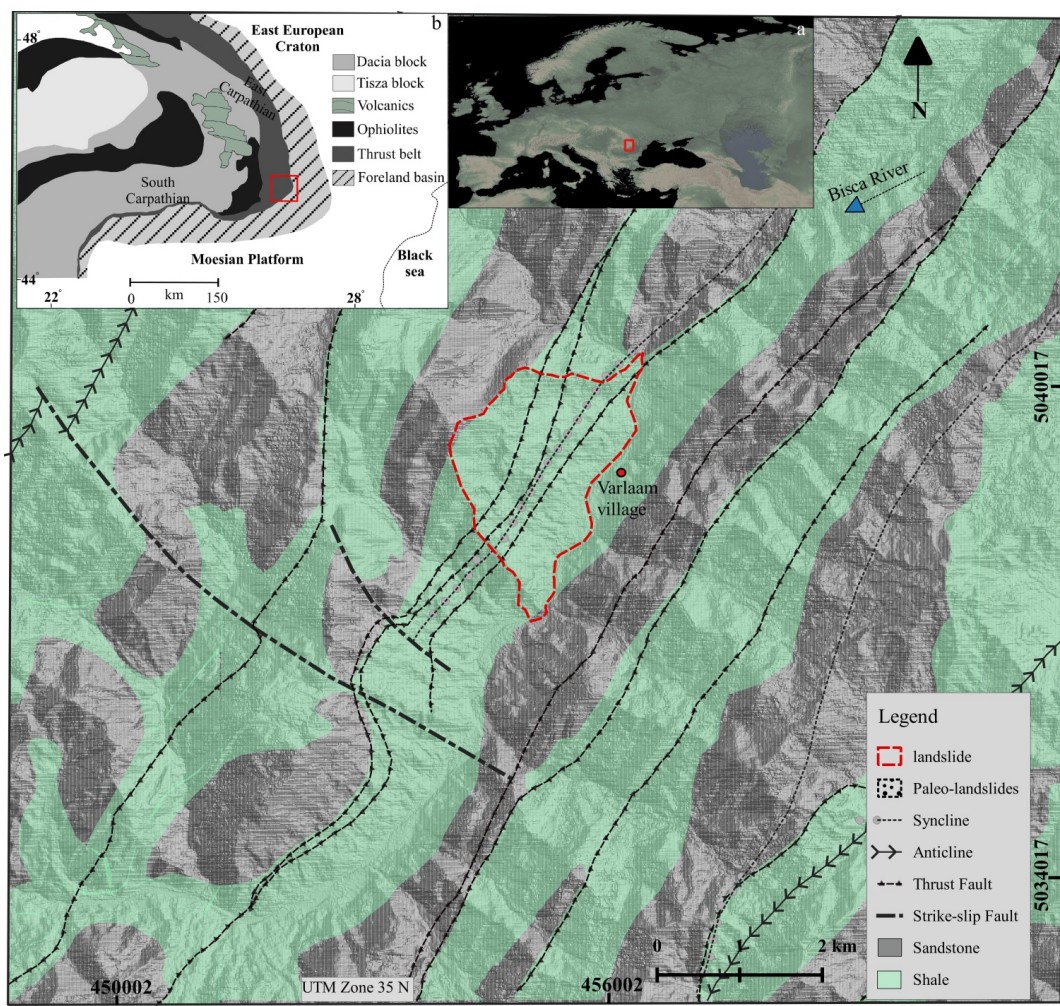


**Figure 1:** Study area. Inset 'a' (source: NOAA/NCEI, USA) 'b' (after Ustaszewski et al.
2008) highlight the position of study area. Geological setting and Paleo-landslides locations
are based on Murgeanu et al. 1965; Tischler et al. 2012; Pospisil and Hipmanova 2012.



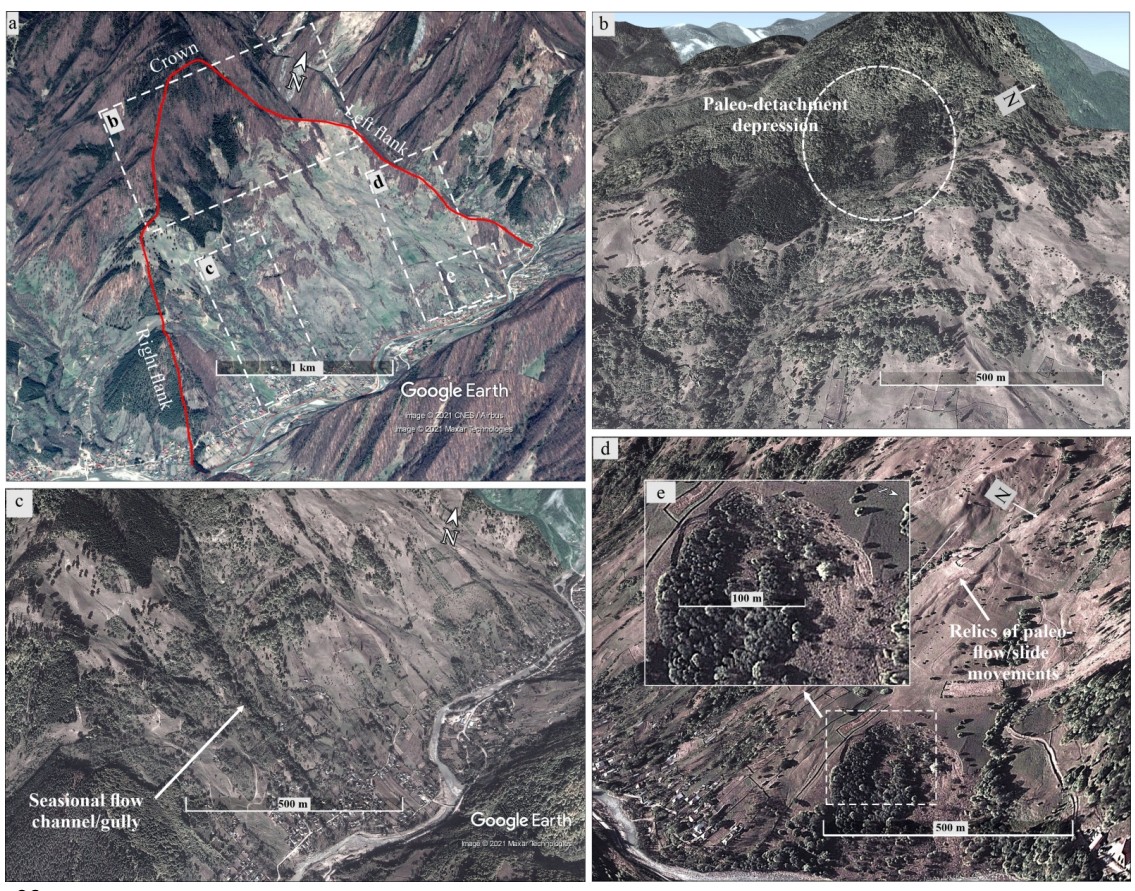


**Figure 2**: Landslide features. (a) Landslide marked with different features, (b) Crown portion, (c) Right flank, (d) Left flank, (e) Signs of failure in the flow deposits. Image Source: Google Earth.

2.2 Rainfall and earthquake regime

The average annual rainfall in the region has been 756± 120 mm/yr during the years 2000-2019 (Fig. 3). This uncertainty of ± 120 mm/yr in average annual rainfall is referred to relatively higher annual rainfall in the last decade particularly in the years 2010, 2013, and 2016 (Fig. 3a). Monthly rainfall patterns further reveal relatively higher rainfall in the months of May, June, and September in the last decade (Fig. 3b). Notably, June-September constitute the summer season in the study area. Such enhanced summer rainfall has been related to the existing positive phase of the North Atlantic Oscillation (NAO) index that allows the strengthening of continental climate, Mediterranean retrogressive cyclones, and Siberian High in central and southern Europe (Constantin et al. 2007; Magyari et al. 2013; Obreht et al.




2016). Further, the daily rainfall data of the years 2000-2019 revealed 55 extreme rainfall
events (Fig. 3c). 'Extreme' rainfall pertains to >30 mm/24h in the region on the basis of
previous studies exploring the rainfall variability (Apostol 2008; Croitoru et al. 2016). Out of
these 55 events, 32 events with a total cumulative precipitation of about 1263 mm occurred in
the last decade, particularly in the years 2010, 2013, 2016-2018.

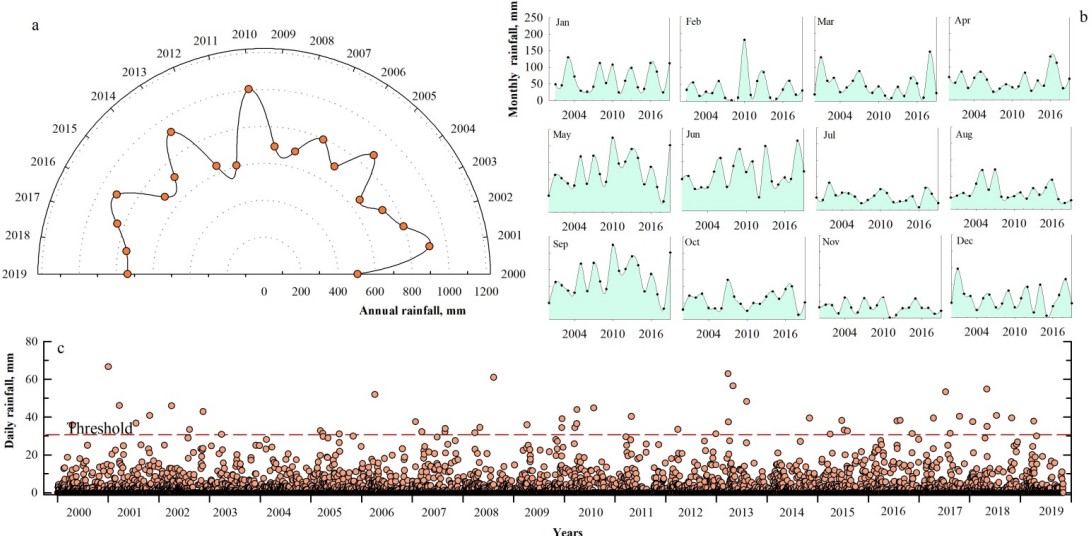


**Figure 3**: Rainfall pattern. (a) Annual variation, (b) Monthly variation, (c) Daily variation.
        Data source: GPM_3IMERGDF v.06 (Huffman et al. 2019). Spatial resolution: 0.1°,
        temporal resolution: daily. Threshold (or extreme) is based on Apostol (2008);
        Croitoru et al. (2016).

Apart from the rainfall, soil moisture and surface runoff pattern also showed temporal
increase as the annual average of these parameters increased in the years 2010-2019 (Fig. 4a,
b, c). The years 2005 and 2010 witnessed the peaks of all three variables that might be one of
the reasons for the debris flows and flash floods in the region in these years (Micu et al. 2013;
Grecu et al. 2017). The temporal increase of these parameters is also evident in the monthly
regime (Fig. 4d, e, f). Further, the temporal pattern of relatively higher values (above-average)
of rainfall, surface runoff, and soil moisture revealed that May-September months dominate
the trend having majority of the events when all three variables had extremes (i.e., above-
average) (Fig. 4g). These 'above-average' values refer to the monthly scale. This temporal
overlapping of these variables further justifies the occurrence of debris flows and flash floods





in this region in the last decade and possibility of more such events in the near future (Micu et
al. 2013; Ilinca 2014; Grecu et al. 2017; Micu et al. 2019).

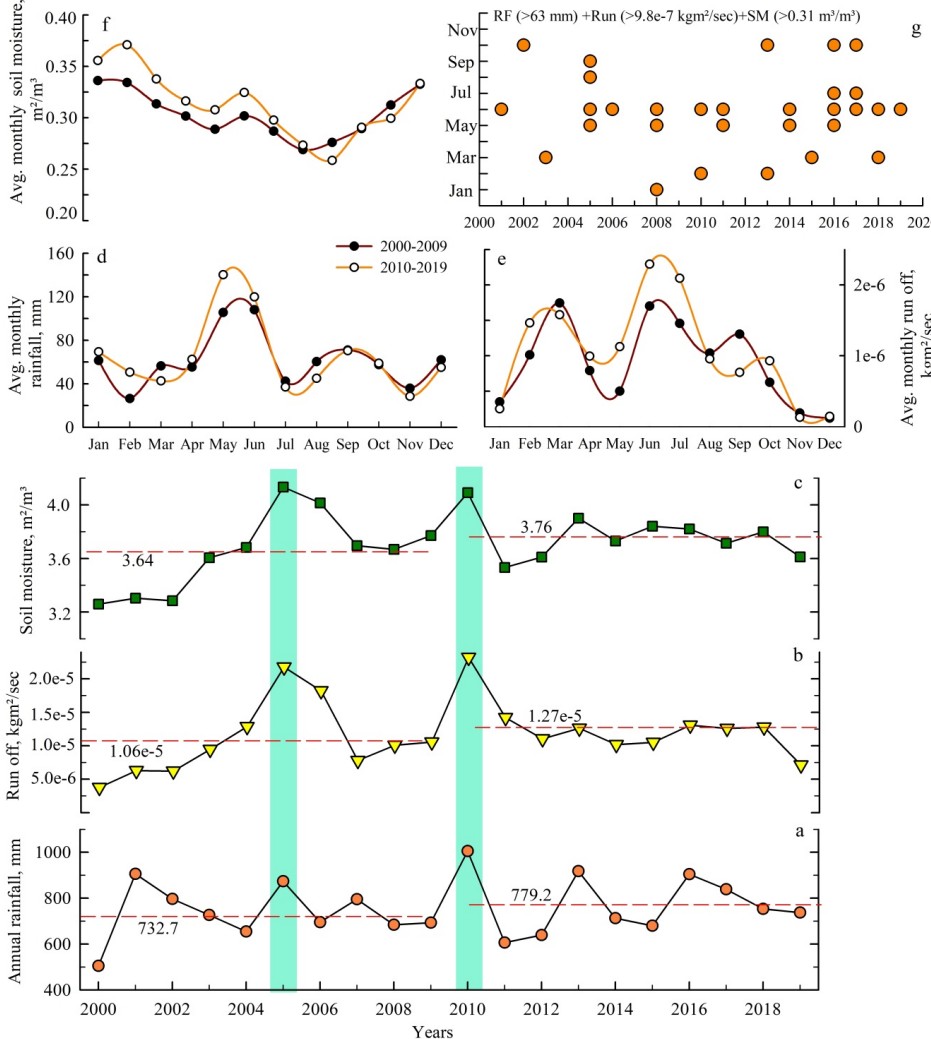

**Figure 4:** Relationship of rainfall, surface runoff, and soil moisture. (a-c) Annual pattern.
Green bars refer to peaks of all three variables in these years. (d-f) Average monthly
pattern, (g) Months having above-average values of rainfall, runoff, and soil moisture.
Data Source: Surface runoff data (FLDAS_NOAH01_C_GL v. 01, McNally et al.
2017). Soil moisture data (GLDAS_CLSM025_DA1_D, Li et al. 2020). Spatial
resolution: 0.1°, temporal resolution: monthly.


Apart from the temporally enhanced rainfall, surface runoff, soil moisture, the study area is
also subjected to frequent earthquakes owing to its position in the Vrancea Seismic Zone that
is one of the most active seismic zones in Europe (Fig. 5).

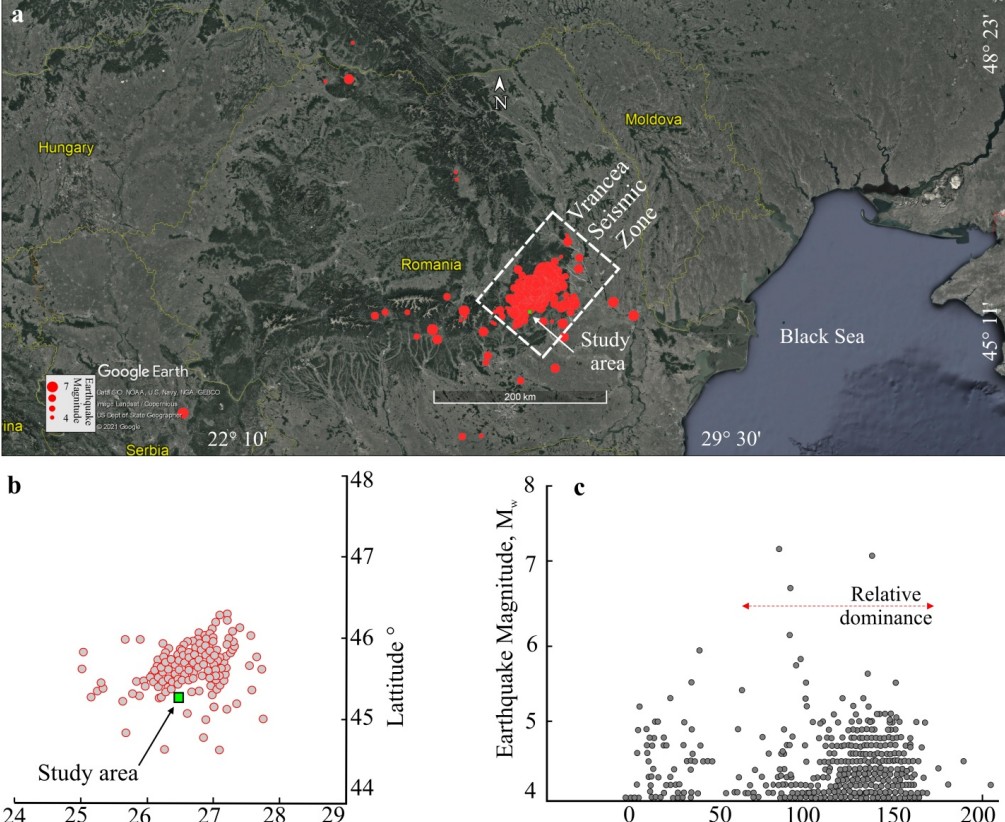


**Figure 5:** Earthquake pattern. (a-b) Position of study area (c) Depth and Earthquake
magnitude. Data source: National Institute for Earth Physics, Romania.

This region has received ~490 earthquakes ($M_w \geq 4$) during the years 1960-2019. The
earthquake event cluster represents a NE-SW trend (Fig. 5b). About 75 % of the total
earthquake events occurred in a depth range of 60-180 km (sub-crustal depth) and 4 out of 5
events having a magnitude $\geq 6$ occurred within 60- 100 km depth (Fig. 5c). The relative
dominance of $M \geq 6$ earthquakes in this depth range has been related to the reverse faulting
mechanism in this depth range (Radulian et al. 2007; Petrescu et al. 2019). The possible





explanation of the pattern of earthquakes has been divided in the following two
categories; (1) it might be associated with descending relic ocean lithospheric beneath the
bending zone of the SE Carpathians , or (2) it might be associated to continental lithosphere
that has been delaminated, after the collision (Bokelmann and Rodler, 2014; Petrescu et al.
2019). These frequent earthquakes in the region have caused many landslides and any major
future earthquake might have ground effects in a much larger area (150000 km$^2$), possibly
causing more landslides (Havenith et al. 2016).
**3 Methodology**
In order to evaluate the landslide response under seismic and extreme rainfall conditions, our
approach involved data collection from field and numerical simulations (slope stability and
runout analysis). Details are as follows;
3.1     Debris (or loose material) depth estimation
We analysed seismic ambient noise at 56 measure points to estimate the depth of impedance
contrasts. The equipment was composed of 7 velocimeters Güralp CMG-6TD 30s and 1
velocimeter Lennartz 5s and Cityshark II. The technique aims at estimating the site resonance
frequency by computing the spectral ratio between horizontal (NS, EW) and vertical
components (Nakamura, 1989). Under particular geological conditions where impedance
contrast exists at depth, as representative of a loose/soft material overlying bedrock, the
resulting Horizontal to vertical spectral ratio (HVSR) curve presents a peak in correspondence
of the site resonance frequency ($f_o$). Fig. 6a represents the location of the inferred $f_o$ in a range
of <1.5-4.5 Hz. Lower frequencies, generally implying relatively higher thickness of loose
material, are noted in the central part and near the right flank.
The thickness (h) of the loose/soft material is consecutively estimated using the shear-wave
velocity ($V_s$) and resonance frequency ($f_o$) using the following equation (Murphy et al. 1971;
Ibs-von Seht & Wohlenberg 1999);
$$h = V_s / (4 * f_o) \qquad \textbf{Eq. 1}$$
In view of the similar litho-tectonic conditions and spatial proximity, the shear-wave velocity
($V_s$) values in the present study are based on Mreyen et al. (2021). For the loose overburden
(soil) and rockmass, the $V_s$ are taken as ~400 m/sec and ~900 m/sec, respectively.




The thickness of the loose material (inferred from the HVSR and $V_s$) at different measurement
locations was later imported in the LeapfrogGeo software (v. 5.1) along with the surface
morphology (Fig. 6b). The surface morphology with a spatial resolution of 12 m is based on
the TanDEM-X (TerraSAR-X add-on for Digital Elevation Measurement) digital elevation
model. The surface morphology and depth information of loose material were integrated using
the LeapfrogGeo (v.5) to construct a continuous soil thickness layer and hence a 3D model of
the landslide (Fig. 6c, d). This model was later used to extract the 2D slope sections (CS-1,
CS-2, CS-3, and CS-4) for the slope stability evaluation (Fig. 7a).

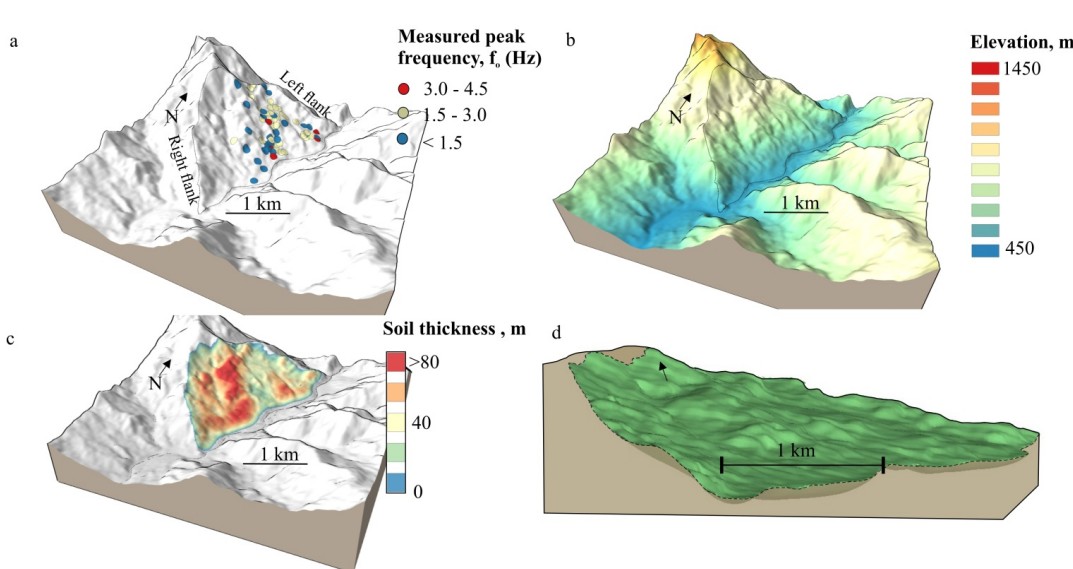


**Figure 6**: Landslide model construction. (a) Measured peak frequency distribution. Based on
Cauchie et al. 2019, (b) Digital elevation model, (c) Soil (or debris) thickness pattern in the
landslide, (d) Cross sectional view of landslide model.

3.2     Slope Stability evaluation
The 2D slope sections (CS-1, CS-2, CS-3, and CS-4) were used to determine the hillslope
response under static (gravity) and dynamic (seismic) conditions by performing the slope
stability analysis in the UDEC v.6 (2014) software. The configuration of these 2D sections is
presented in Fig. 7. Each slope section comprises loose overburden (soil) over rockmass and
an interface joint separating these blocks.

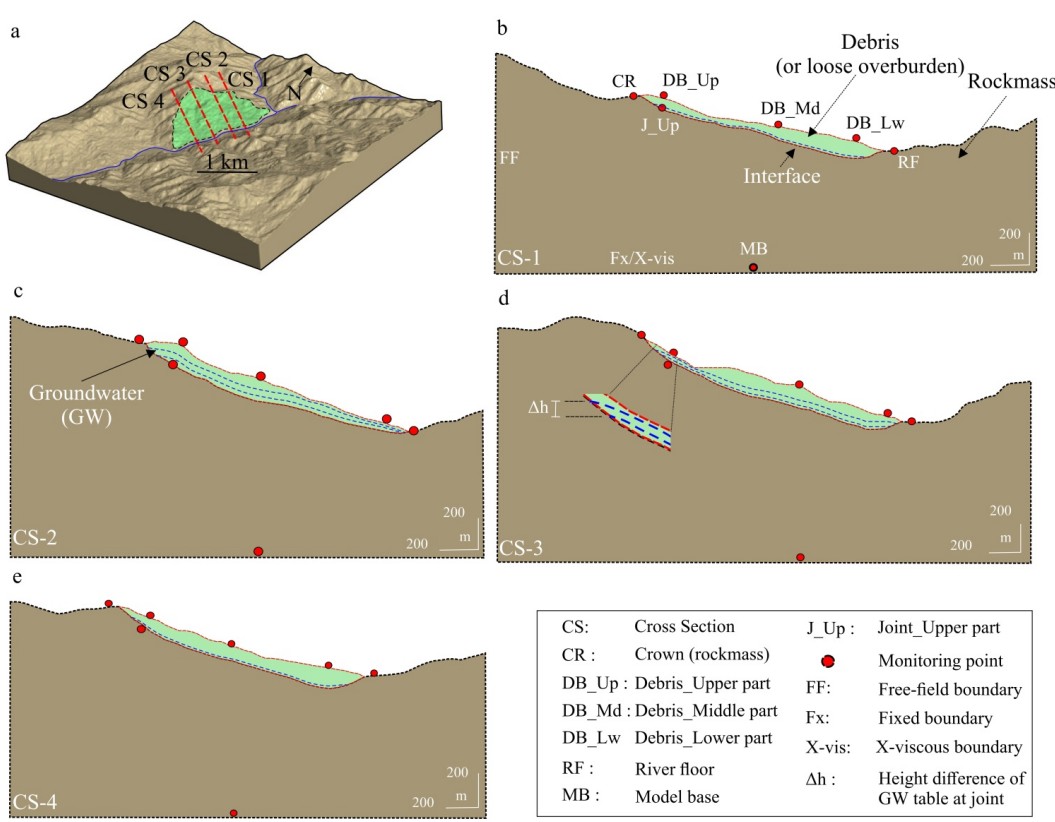


**Figure 7:** Model configuration for the Slope stability analysis. (a) Landslide model. The
location of the different cross sections used in the UDEC models are marked by red lines, (b-
e) Configuration of the sections; CS-1 to CS-4.

Under static condition, factor of safety of slope and potential material displacement are
determined, whereas under dynamic condition, potential material displacement, Peak Ground
Acceleration (PGA), and spectral ratio are evaluated. For the PGA and spectral ratio, material
models are considered as elastic, whereas for the factor of safety and material displacement
(static/dynamic) calculations, elasto-plastic models are considered. Elastic material model
involved modulus (elastic/shear/bulk) values of the rock mass and soil. In elasto-plastic
conditions, Modified Hoek-Brown (MHB) plasticity criteria (Hoek et al. 2002) and Mohr-
Coulomb (M-C) plasticity criteria (Coulomb 1776; Mohr 1914) are used for the rock mass
and soil, respectively. The joint plane is assigned Coulomb-Slip criteria (Coulomb 1776) in


both elastic and plastic conditions. For dynamic analysis, two different signals, i.e. Ricker
wavelet (Ricker 1943) and a signal record of the 1976 Friuli Earthquake, are used (Fig. 8).

**Figure 8**: Seismic signals. (a)
Ricker Wavelet (as recorded at
the model base monitoring
point) (b) 1976 Friuli
Earthquake, (Italy). Note:
Different time scale.

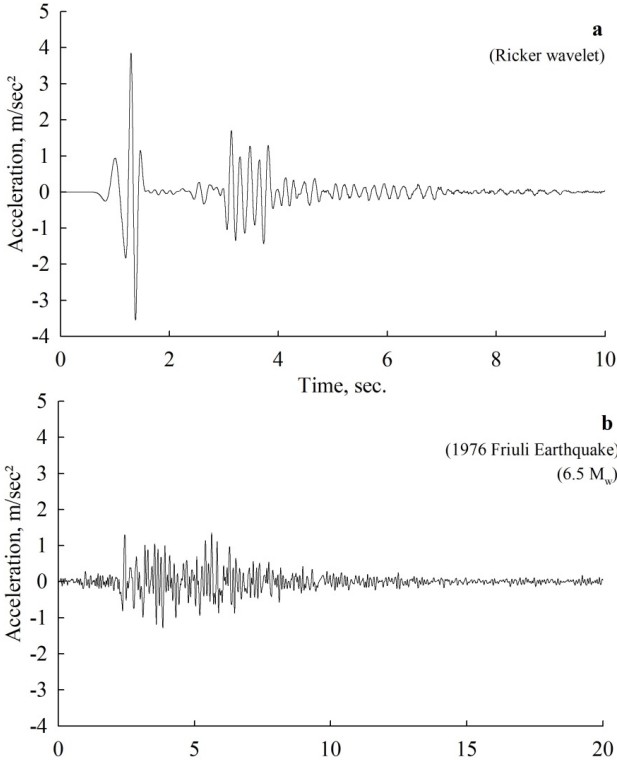

The Ricker wavelet, a theoretical waveform, provides an advantage to be a relatively short
signal marked by an energy distributed over a range of frequencies. Therefore, the PGA and
spectral ratio are evaluated using the Ricker wavelet to understand the ground motion
amplification on the landslide surface. Notably, in many studies such ground motion
amplification is found to enhance the slope instability (Lenti and Martino 2012; Gaudio et al.
2014). The Ricker wavelet has been used in several studies owing to its reliable representation
of seismic waves propagating through the viscoelastic homogeneous media (Bourdeau et al.
2004; Gholamy and Kreinovich 2014). Further, the displacement is determined using both
dynamic signals (Ricker wavelet and Friuli earthquake, 1976) to evaluate the difference.





Soil and rock mass blocks in the sections (CS-1 to CS-4) were discretized into finite
difference zones of 6m and 20m size, respectively according to the following relation
(Kuhlemeyer and Lysmer, 1973);

$\Delta l \leq \lambda/10 \text{ or } \leq \lambda/8$                **Eq. 2**

Here, $\Delta l$ = zone size, $\lambda$ = wavelength associated with the dominant frequency. '$\lambda$' can be
determined using $\lambda= C/f$, where C is the speed of wave propagation associated with the
fundamental frequency (f). For the 'C' (or shear wave velocity) of soil and rock mass, we
used 400 m/sec and 900 m/sec, respectively (sec. 3.1). The 'f'=2.0-4.5 Hz was considered as a
central frequency range. The boundary conditions were fully restrained (base) & X-restrained
(lateral) under static load and free field (lateral) & fixed/X-viscous (base) under dynamic load
(Fig. 7). To approximate the natural attenuation in the models during the seismic loading,
Rayleigh damping with a 0.02 damping ratio (i.e., 2% fraction of critical damping and 2.5 Hz
central frequency was used with the both mass and stiffness damping. Though most of the soil
types and rock mass possess the damping in the 2%-5% fraction of the critical damping
(Biggs 1964), plasticity models (M-C criteria) and presence of joints result in further energy
loss (UDEC v.6 2014). Therefore, the damping ratio was kept at the lower level of the
suggested range.
Since, the area is subjected to temporally enhanced rainfall (sec. 2.2) and some studies have
noted the percolation of rainfall water in the loose material resulting in the Groundwater
(GW) level increase and subsequent slope instability (Van Asch et al. 1999; Liang 2020),
effect of the GW was also explored. The GW was included in static as well as in dynamic
analysis in plasticity conditions. The UDEC allows the GW simulation through the joints as
per the parallel plate model (Witherspoon et al. 1980). The parameters and their values used
in the static and dynamic analysis are mentioned in Table 1.
Table 1: Input parameters and their values used in the static and dynamic analysis.

| Rockmass parameters | values | Rockmass-soil interface (joint) parameters | values | Soil parameters | value |
|---|---|---|---|---|---|
| Density, $\gamma$ (kg/m$^3$) | 2500 | [4]Normal Stiffness, $k_n$ (MPa/m) | 10000 | Density, $\gamma$ (kg/m$^3$) | 1900 |
| [1]Uniaxial Compressive Strength, $\sigma_{ci}$ (MPa) | 30 | Shear Stiffness , $k_s$ ($k_n$/10) | 1000 | [2]Poisson's Ratio | 0.43 |





| [2]Poisson's Ratio | 0.4 | [5]Cohesion, c (MPa) | 0.01 | [2]Young's Modulus, E (MPa) | 869 |
|---|---|---|---|---|---|
| [2]Young's Modulus, E (MPa) | 5670 | [6]Friction angle, Ø | 30° | [2]Bulk Modulus, K (MPa) | 2070 |
| [2]Bulk Modulus, K (MPa) | 9450 | [7]Residual aperture at high stress, m | 0.0001 | [2]Shear Modulus, G (MPa) | 304 |
| [2]Shear Modulus, G (MPa) | 2025 | [7]Aperture for zero normal stress, m | 0.0005 | [5]Cohesion, c (MPa) | 0.01 |
| [3]GSI | 30 | Water density, Gg/m³ | 0.001 | [5]Friction angle, Ø | 28° |
| [3]Material Constant ($m_i$) | 17± 4 | [7]Joint permeability, (1/MPa*s) | $10^8$ | | |
| $m_b$ | 1.3954 | [1]It was inferred from the empirical equation of Kahraman (2001) using the Vs and Vp data of Mreyen et al. (2021). [2]These values were inferred from the empirical equations of McDowell (1990) using the P & S wave velocity of Mreyen et al., (2021). [3]Based on Hoek and Brown (1997) and field observation. [4]It was inferred from from the empirical equations of Barton (1972); Hoek and Diederichs (2006) using the elastic modulus of rock and approximated spacing of joint sets of~5-10cm. This spacing was assumed in view of highly sheared nature of rockmass. [5]Based on Bednarczyk (2018); Peranić et al. (2020) due to similar litho- tectonic conditions. [6]Based on Barton and Choubey (1977). [7]Based on UDEC v.6 (2014). | | | |
| s | 0.004 | | | | |
| a | 0.5223 | | | | |
| [3]D | 0 | | | | |

A parametric analysis was also performed to justify the selection of values of different input
parameters by evaluating the change in the output parameters in response to the change in
different input parameters. Out of four slope sections, the CS-2 and CS-3 were chosen to
perform the parametric analysis in view of their central position in the landslide and the
heterogeneity in soil thickness and topography (Fig. 7c, d). In order to understand the effect of
the GW level change, two GW levels were considered in the CS-2 and CS-3 sections. Since
the UDEC simulates the fluid flow through joint aperture, the GW level change is manifested
by different heights (h1, h2) of the GW at the joint. Here, the difference of h1 and h2 i.e., Δh
is 10m (Fig. 7d). Among the different input parameters listed in Table 1, angle of internal
friction of soil, joint friction angle, groundwater head, and elastic modulus were used for the
parametric analysis. It is to note that the bulk and shear modulus were also changed along
with elastic modulus because all three modulus parameters are interrelated (Mc Dowell 1990).
Though each parameter might have a certain effect on the output, these four have been noted
to affect the Factor of Safety and displacement relatively more (Kumar et al. 2021).



3.3   Run-out simulation
The hillslopes affected by the seismic shaking have also been noted to be more prone to
rainfall induced slope failures, particularly in the form of debris flows (Shieh et al. 2009;
Tang et al. 2011). Such debris flows can initiate either by increased pore pressure or runoff
involving entrainment (Godt and Coe 2007). Thus, the increased frequencies of the extreme
rainfall, soil moisture, surface runoff, and recent debris flows events in the region (sec. 2.2),
escalate the possibility of debris flow in the Varlaam landslide.
To ascertain the outreach of such potential debris flow during an extreme rainfall event,
Voellmy friction law based model was simulated using the Rapid Mass Movement Simulation
(RAMMS) software. The RAMMS divides the frictional resistance into a dry-Coulomb type
friction ($\mu$) and viscous-turbulent friction ($\xi$) (Christen et al. 2010). The frictional resistance S
(Pa) is thus;
$$S = \mu N + (\rho g u^2)/\xi \qquad \text{Eq. 3}$$
Where $N = \rho hg\cos(\phi)$ is the normal stress on the running surface, $\rho$= density, g= gravitational
acceleration, $\varphi$= slope angle, h= flow height and u= $(u_x, u_y)$, consisting of the flow velocity in
the x- and y-directions.
Generally, the values for $\mu$ and $\xi$ parameters are achieved using the reconstruction of real
events through simulation and subsequent comparison between dimensional characteristics of
real and simulated event. However, the toe of Varlaam landslide merges with the river floor
and hence there is an uncertainty in reconstruction of the volume of previous flow events that
has been washed away by the river. Therefore, $\mu$ and $\xi$ are taken in view of topography of
landslide slope and run-out path, landslide material, and based on previous studies/models
(H¨urlimann et al. 2008; Rickenmann and Scheidl 2013; RAMMS v.1.7.0). In this study,
maximum allowable friction ($\mu$) i.e., $\mu$= 0.4 (or $\phi$ = 21.8°) was used with the turbulence ($\xi$) of
250 m/sec$^2$ (Table 2).
Table 2: Details of input parameters for run-out analysis.

| Landslide | Material type[1] | Material depth[2], m | Friction coefficient[3] | Turbulence coefficient[4], m/sec$^2$ |
|---|---|---|---|---|
| Varlaam | Clayey Silt | 5, 10, 15, 20 | $\mu$= 0.4 | $\xi$ = 250 |

[1] Field based approximation. [2] Considering that fact that during slope failure, irrespective of type of trigger, entire loose
material might not slide down, the depth is taken as a variable. [3] In order to keep the results of conservative nature & presence
of vegetation, we have taken a maximum allowable friction i.e., $\mu$= 0.4 (Hungr et al., 1984; RAMMS v.1.7.0). This case is



considered to understand the potential impacts of debris flow even after the maximum friction. [4]This range is used in view of
the type of loose material i.e., cohesive (RAMMS v.1.7.0).
**4 RESULTS & DISCUSSION**
4.1 Slope stability evaluation
4.1.1  Factor of Safety (FS) & displacement
The FS of slope varies in a range of 1.17-1.32 that decreases further to 1.09-1.29 under
Groundwater (GW) condition (Fig. 9). In both cases, the CS-2 model attains lowest FS
implying relatively more instability. The displacement in loose material was obtained in
static, static with fluid (GW), dynamic, and dynamic with fluid (GW) conditions.  Under the
static condition, displacement ranges between 0.4-4.0 m that increases to 0.68 m-18 m under
the GW condition with minimum at CS-1 and maximum at CS-2 (Fig. 9).  Under dynamic
condition, displacement ranges from 8-60 m, and further increases to 7.5-62 m by combining
dynamic with GW conditions. Similar to the static condition, minimum displacement is noted
at CS-1, whereas maximum at CS-2. Further, in all sections (CS-1 to CS-4), displacement
accumulated mostly at the upper part of the debris layer (i.e., landslide crown) or at the
steepest portion of slope surface. This spatial affinity of displacement and steep gradient is
caused by the influence of topography on the material displacement (Kumar et al. 2021). It is
to note that this dynamic displacement pattern pertains to the Friuli earthquake signal (Fig.
8b). A comparison of the static and dynamic displacement (caused by the Friuli earthquake
signal and Ricker wavelet) is presented in Fig. 10.
As also shown in Fig. 9, the GW condition enhanced the displacement in static as well as in
dynamic conditions (Fig. 10). Static displacement showed least scattering as evident from the
median level and least difference of Max. and Min. values. Further, except for the CS-2
section, all three sections (CS-1, 3, 4) have relatively low dynamic displacement in dry and
wet (GW) conditions due to the Ricker wavelet than compared to the displacement caused by
the Friuli signal (Fig. 10a-d). This difference may be attributed to the response of steep
topography (of CS2 model) to the multi-frequency signal (Ricker wavelet).




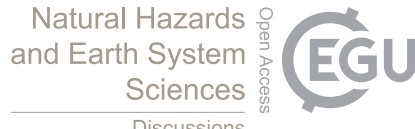




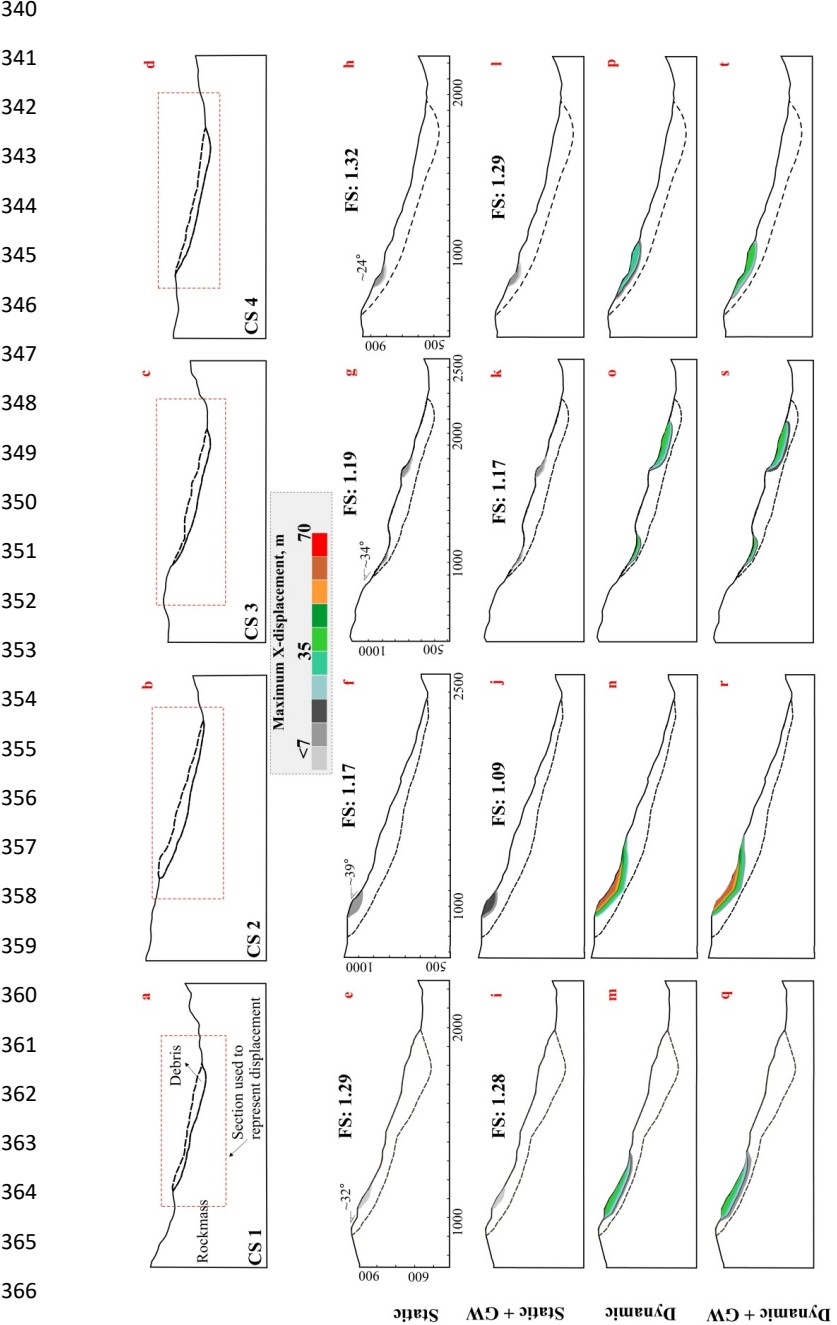

**Figure 9**: Factor of Safety (FS) and material displacement (X-direction). a – d refer to original slope sections with sub-sections (red rectangle) used to represent displacement. e –h, i-l, m-p, and q-t refer to displacement in static, static +GW, dynamic, and dynamic +GW conditions, respectively.

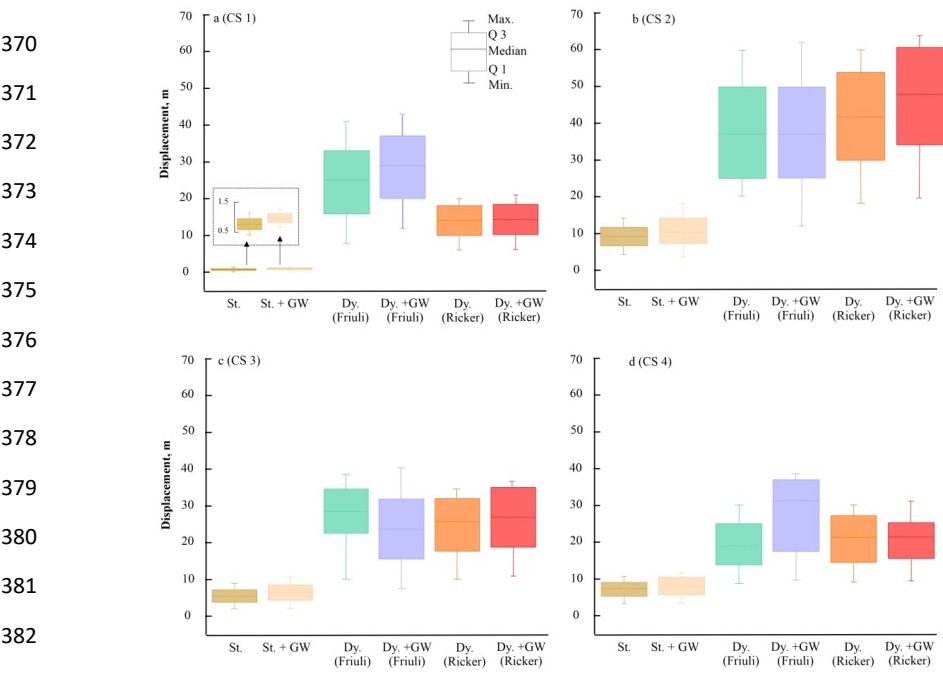

**Figure 10**: Comparison of material displacement under different conditions. St. and Dy. Refer to Static and Dynamic conditions, respectively. GW refers to Groundwater.

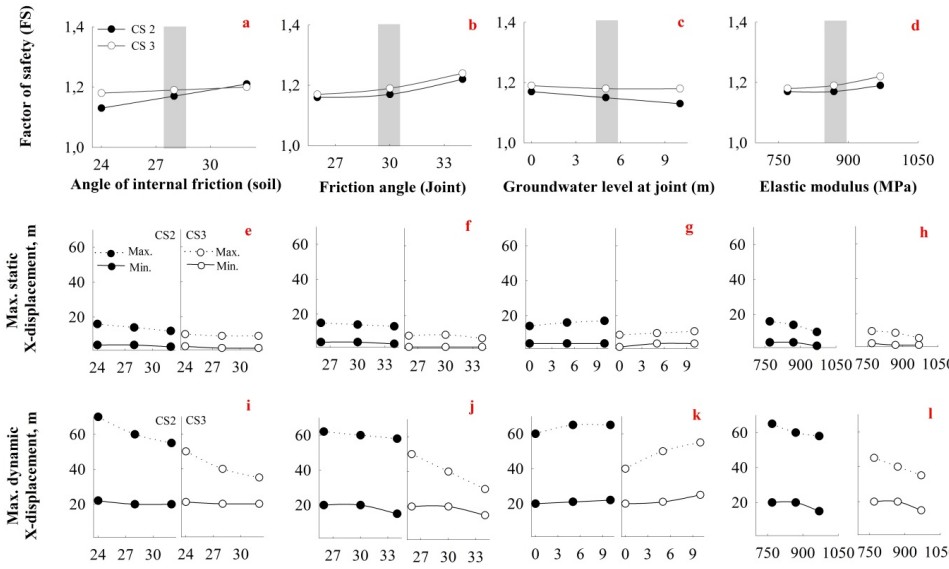

**Figure 11**: Parametric analysis. (a-d) Variation in the FS, (e-h) Variation in the static displacement, (i-l) Variation in the dynamic displacement. Grey bar represents the values that are used in the slope stability analysis (sec. 4.1.1).




4.1.2 Parametric analysis
The Factor of Safety (FS) of slope increased in response to increase in angle of internal
friction of soil, joint friction, and elastic modulus (Fig; 11). Relatively higher increase in the
FS (~7% in the CS 2) is attained by increasing the angle of internal friction of soil. This effect
is attributed to the 'Shear Strength Reduction (SSR)' approach (Matsui and San 1992;
Griffiths and Lane 1999) that was used to determine the FS. The GW level increase resulted
in a decreasing FS because the increased GW level increased the joint flow rate, as per
'Parallel-Plate model' (Witherspoon et al. 1980), and thus enhanced the fluid pressure on the
overlying medium i.e., soil. This increased fluid pressure further decreased the normal stress
and hence the shear stress of the overlying soil, as per Mohr's Criteria (Mohr 1914). Such
decrease in the shear stress of soil resulted in the decreased FS.
Since material displacement is a spatially variable parameter, as shown in Fig. 9, Static and
dynamic displacements are represented in a range of maximum (max.) and minimum (min.) in
Fig. 13. Static displacement is noted to decrease on increasing the angle of internal friction of
soil, joint friction, and elastic modulus. Relatively higher decrease (~40% in CS 2 and ~38 %
in CS3) occurred in response to the modulus increase. This decrease in the displacement is
referred to fact that increased modulus increases the normal and shear strength of the soil and
hence displacement will decrease on increasing the modulus (Hara et al. 1974). The GW level
increase resulted in the increased static displacement (~16% in CS2, ~36% in CS3). Such
increase in the static displacement is attributed to the decreased shear strength of soil due to
the increased joint fluid pressure (Witherspoon et al. 1980).
Similar to the static displacement, dynamic displacement decreased on increasing the angle of
internal friction of soil, joint friction, and elastic modulus and increased on increasing the GW
level. Along with the modulus, angle of internal friction of soil is also noted to decrease
(~16% in the CS2, ~21% in the CS 3) the dynamic displacement relatively more. The increase
in the GW level resulted in 8% and 33% increase in the CS2 and CS3 models in dynamic
displacement.
Notably, present study utilized approximated values of the input parameters for the slope
stability analysis (Table 1). Though approximated values cannot replace the values measured
in the geotechnical analysis, parametric analysis minimizes the uncertainty caused by
selection of specific values by exploring the possible output pattern.





Thus, aforementioned findings of the parametric analysis highlight the potential uncertainty in
the FS and material displacement (static/dynamic) that can arise due to the input values. By
utilizing the central values (highlighted as grey) in the slope stability findings (sec. 4.1.1), the
present study attempted to minimize such uncertainty in the findings. Further, though the GW
was also used in the UDEC models to infer the influence of saturation on slope stability,
potential response of the slope under excessive saturation (extreme rainfall) is further
explored through the runout prediction (sec. 4.2).

### 427      4.1.3    Peak Ground Acceleration (PGA)

Apart from the FS and displacement, ground motion (acceleration) amplification was also
evaluated to understand the potential seismic deformation at the slope surface. The input
seismic signal for the following acceleration pattern is presented in Fig. 8a. For all four
models (CS1 to CS4), the PGA values at the river floor (RF) ranges between 5.78-7.47 m/sec²
(0.58g – 0.74g), whereas at the rock mass surface above the landslide crown (CR) it varies
from 6.37 to 10.19 m/sec² (0.65g -1.03g) (Fig. 12). At the model base (MB), maximum
acceleration remains between 3.79-3.90 m/sec² (0.38g -0.39g).
Thus, the PGA at the river floor (RF) amplifies~1.5-2.0 times from the maximum acceleration
at the model base, whereas at the rock mass surface above the landslide crown, it amplifies
~1.7-2.7 times from the maximum acceleration at the model base. Such amplification of the
PGA at the rock mass surface above the landslide crown can be attributed to the topographic
irregularity and upward propagation of seismic waves where they meet preceded waves
produced on the relatively horizontal surface of the slope (Jibson 1987; Havenith et al. 2003;
Bourdeau and Havenith 2008; Luo et al. 2020).
The debris surface, however, attains relatively higher PGA in all four models than the rock
mass surface as noted at the following three monitoring stations; DB_Lw, DB_Md, and
DB_Up (Fig. 12). At the lower part of the debris (DB_Lw), the PGA ranges from 8.3 to
12.13 m/sec² (0.84g-1.23g) that further grew at the middle part of the debris (DB_Md) and
attaines10.17-14.40 m/sec² (1.03g-1.46g). The maximum PGA is attained by the upper part of
the debris (DB_Up) with a range of 7.26-18.50 m/sec² (0.74g - 1.88g). Such relatively high
PGA at the debris surface can be referred to the impedance contrast between underlying rock
mass and overlying soil and/or partial loss of the shear strength during seismicity (Novak and
Yan, 1990; Safak, 2001).


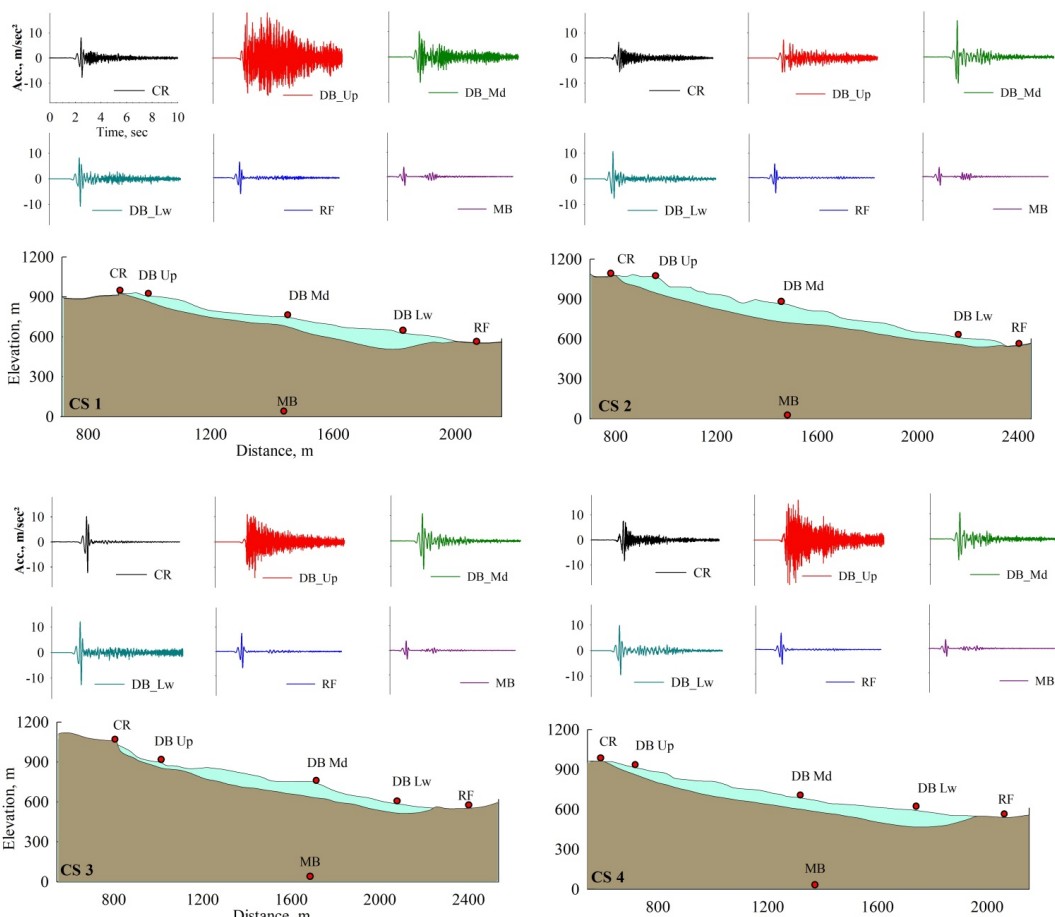

**Figure 12**: Maximum acceleration at different monitoring points. CR: Crown (Rock mass), DB Up: Debris upper part, DB Md: Debris middle part, DB Lw: Debris lower part, RF: River Floor, MB: Model base.

Detailed evaluation at different monitoring points in each model are as follows; Model Base (MB) and River floor (RF) monitoring points have almost similar maximum acceleration values in all four models. At the lower part of the debris i.e., DB_Lw, relatively higher PGA is attained by the CS3 model (~12.1 m/sec²) followed by the CS2 model (~10.8 m/sec²) in comparison to DB_Low points of CS1 and CS4. Relatively higher PGA is attributed to lower soil thickness below this monitoring point in the CS3 and CS2 models that could be the main





reason for acceleration amplification as also stated by Murphy et al. (1971); Beresnev and
Wen (1996).
At the middle part of the debris i.e., DB_Md, relatively higher PGA is attained by the CS2
model (~14.4 m/sec²). Notably, despite the relatively higher soil thickness, this monitoring
point obtained a relatively higher PGA. It possibly occurred due to irregular topography of the
CS2 model that generally results in interference of direct and scattered waves and hence
amplification of ground motions (Asimaki and Mohammadi 2018).
At the upper part of the debris i.e., DB_Up, relatively higher PGA is attained by the CS1
model (18.5 m/sec²) followed by the CS4 model (15.8 m/sc²). The effect of soil thickness
below this monitoring point, as explained for the lower part of debris, could be the main
reason for such amplification at this monitoring point in these models. Monitoring point at
rock mass surface above the landslide crown (CR) too has almost similar PGAs in all the
models except the CS3 model. Relatively higher PGA (10.19 m/sec²) at the CR monitoring
point of CS3 model might be due to its position on steeper surface, whereas CR points at
other models are at relatively flat surface.
4.1.4   Spectral Ratio
The ground motion amplifications were also explored using the spectral ratios at two central
slope sections; CS-2 and CS-3 (Fig. 13). In both models, the (River Floor) RF point showed
no significant amplification at any particular frequency, possibly due to the flat surface
positioning. In CS2 model, Debris Lower part (DB_Lw) point shows notable amplification at
2.0-2.5 Hz with minor amplification at 4.5-5.0 Hz, whereas in the CS 3 model, DB_Lw point
shows attenuation (or de-amplification) near ~2 Hz and slight amplification at 4.5-6.0 Hz. The
contrast of amplification and de-amplification at ~2 Hz is attributed to the geometrical
variation in topography because the DB_Lw point in the CS2 is situated at a relatively
elevated surface, whereas in the CS3, at a relatively shallow surface. Minor geometrical
variations at the slope toe have been observed to result in de-amplification at low frequencies
in other studies also (Bouckovalas and Papadimitriou, 2005).
Notably, along with the DB_Lw point, Debris Middle part (DB_Md) and Debris Upper part
(DB_Up) points in both the models also have minor/major amplification at 4.5-6.0 Hz. This
coexistence of amplification at a certain frequency range by different monitoring points at
debris surface may be attributed to impedance contrast between debris and underlying rock




mass. Further, the DB_Md point in both the models showed amplification at ~1.0 Hz and 2.0-
2.5 Hz. The amplification at lower frequency i.e.,~1.0 Hz may be attributed to the thick (40-
60m) layer of debris that possibly decreases the resonance frequency and results in
amplification of ground motion as also reported by Beresnev and Wen (1996). The
amplification at 2.0-2.5 Hz may be referred to the elevated topography at these points in both
the models.

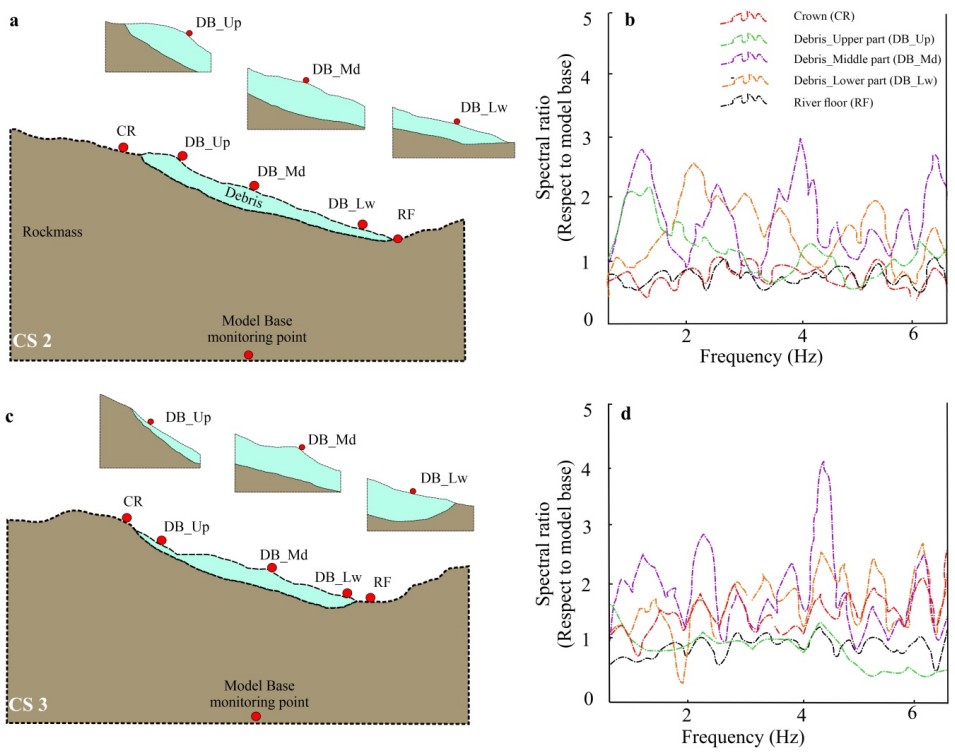


**Figure 13**: Spectral ratio pattern. (a) CS-2 model with the position of monitoring points and
zoomed regions of debris monitoring points. (b) Spectral ratio pattern in the CS-2, (c)  CS-3
model with the position of monitoring points and zoomed regions of debris monitoring points,
(d) Spectral ratio pattern in the CS-3.

The DB_Up point in both the models has different responses. In the CS2 model, it showed
amplification at 1.0-1.5 Hz, whereas in the CS3 model, spectral ratio is relatively stagnant
except minor amplification at 4.0 & 6.0 Hz. This contrast may be understood by the fact that
in the CS2, this monitoring point is situated at a thicker and elevated surface, whereas in the


CS3, it is situated at relatively shallow topography and on top of relatively thin landslide
thickness.
Finally, the Crown (CR) point also has a different spectral ratio in both the models. It shows
higher amplification in the CS3 model than the CS2 model that may be referred to the
positioning of these points. The CR in the CS2 is situated at a relatively flat surface unlike in
the CS3 model where it is situated at a steep surface. Thus, the monitoring points showed
amplification at multiple frequency range that is attributed to complex topography of
landslide, soil thickness variation, and impedance contrast.
4.2 Landslide runout pattern
In view of uncertainties to ascertain the exact depth of loose material that will be
eroded/entrained during the debris flow, runout pattern was evaluated at four different depths;
5m, 10m, 15m, and 20m of the loose overburden (Fig. 14a, b). Runout characteristics (flow
height/flow velocity) of the debris flow that will strike the river floor during such an event are
also inferred along the river channel (Fig. 14c).
At 5 m soil thickness, the landslide resulted in a maximum flow height of ~8 m and maximum
flow velocity of ~4.5 m/sec (Fig. 14 d, e). Along the river channel, flow attained a maximum
height of ~9 m near the right flank and maximum velocity of ~3 m/sec near the left flank of
the landslide (Fig. 14f). At 10 m soil thickness, the landslide resulted in a maximum flow
height of ~20 m and maximum flow velocity of ~10 m/sec (Fig. 14 g, h). Along the river
channel, flow attained a maximum height of ~16 m and maximum velocity of ~2.9 m/sec near
the right flank (Fig. 14i). At 15 m soil thickness, the landslide resulted in a maximum flow
height of ~30 m and maximum flow velocity of ~16 m/sec (Fig. 14 j, k). Along the river
channel, flow attained a maximum height of ~22 m and maximum velocity of ~2.2 m/sec.
near the right flank (Fig. 14l). At 20 m soil thickness, the landslide resulted in a maximum
flow height of ~42 m and maximum flow velocity of ~21 m/sec (Fig. 14 m, n). Along the
river channel, flow attained a maximum height of ~26 m and maximum velocity of ~2.1
m/sec near the right flank (Fig. 14o).





**Figure 14**: Debris flow run-out pattern. (a) Soil (or debris) thickness pattern in the landslide,
(b) Different depths (5, 10, 15, and 20 m) used for the analysis in the 60-80 m thickness
region, (c) River profile section A-B used to represent the resultant debris flow runout along
the river, (d-f) results at 5 m depth, (g-i) results at 10 m depth, (j-l) results at 15 m depth, (m-
o) results at 20 m depth.
Further, in order to understand the extent of runout along the river channel, runout results at
maximum considered thickness (i.e., SE=20 m) were also laid over the Google Earth imagery
(Fig 15a, b). A top view of the landslide with the runout is shown in inset 'c'. The predicted
runout is noted to extend across the river channel mainly at two locations, one near the left
flank (Fig. 15d) and the other near the right flank (Fig. 15e). At both of these locations, the
river channel attains sinuosity in a range of ~1.30-1.32 (shown through channel length
measurement). River channel might owe this sinuosity to the paleo-landslide and/or fluvial
deposit that is extending the slope toe at these locations. Thus, the runout findings of present
study are noted to follow the same spatial extent as possibly followed by previous landslide
events.

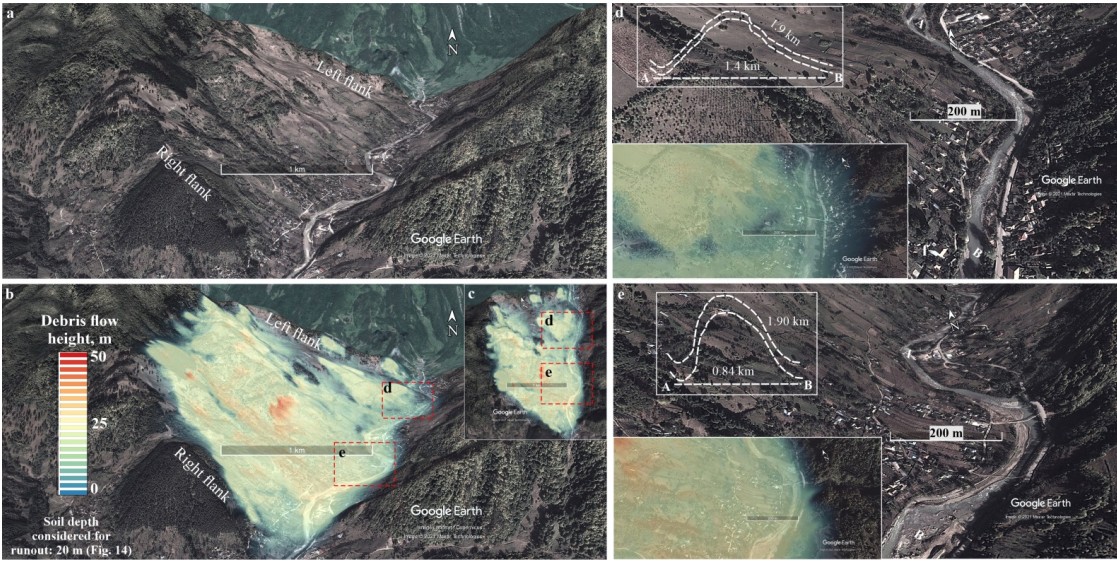

Fig. 15: Debris flow run-out pattern at 20 m depth. (a) Upstream view of landslide from the
right flank, (b) Run-out pattern at 20 m depth, (c) Top view of landslide highlighting
two regions where runout reached across the river (d) Runout pattern near left flank
extending across the river channel, (e) Runout pattern near right flank extending across
the river channel.

**5 SUMMARY**
The present state of slope reveals an instability condition through the Factor of Safety (FS) in
a range of 1.09-1.32 and potential displacement near the landslide crown (Fig. 9, 10). Such a
displacement near the landslide crown has been related to the development of shear failure in
slopes (Matsui and San, 1992; Kumar et al. 2018; 2021). The possibility of shear failure
becomes more viable in case of degradation of shear strength of slope material and/or rupture
planes. Notably, both the main landslide triggering factors; rainfall and earthquake have been
found to degrade the shear strength of slope material through the percolation and shaking
induced particle movements, respectively (Cai and Ugai, 2004; Chang and Taboada 2009).
The GW, implying the rainfall induced percolation effect, further decreases the Factor of
Safety and increases the material displacement (Fig. 9, 10). This effect of the GW is attributed
to the hydraulic pressure in the joint against the overlying loose material that decreases the
normal stress and hence the shear strength of overlying loose material (Mohr, 1914;
Witherspoon et al. 1980).
Similar to the GW effect in static condition, the combined response of the dynamic force and
the GW resulted in an increase of the displacement (Fig. 9). Increased displacement during
the seismic force can be understood from the following equation (Cundall 1980);

$$u = \left( \frac{\int(\sigma.n.ds+F)}{m} \right) + g \qquad \text{Eq. 4}$$
Here, u= displacement, σ= zone stress tensor, s= surface enclosing the mass (m), n= unit
normal to s, g= gravitational acceleration, F= resultant force ( $F^z$+ $F^c$+ $F^e$). $F^z$ = internal stress
in zone, $F^c$ = contact forces between blocks (joint), $F^e$ = external force. Here, seismic force is
represented by the $F^e$.
The enhanced material displacement during the combined effect of the dynamic force and the
GW can be attributed to the fact that seismic shaking increases the hydraulic pressure in the
joints that causes enhanced material displacement in the overlying loose material (Wang et al.

2010).

Apart from the Factor of Safety (FS) and material displacement, ground motion amplification
also revealed slope instability (or potential deformation). The maximum value of Peak





Ground Acceleration (PGA) is attained by the upper part of the debris surface (near the
landslide crown) (Fig. 12) that is referred to the impedance contrast between underlying rock
mass and overlying soil and/or partial loss of the shear strength during seismicity (Novak and
Yan, 1990; Safak, 2001). Further, the spectral ratio also showed signal amplification, at
multiple frequency range, at the debris surface (Fig. 13). Such amplification at multiple
frequency ranges is attributed to complex topography of landslide, soil thickness variation,
and impedance contrast (sec. 4.1.4). Such high amplification at the slope surface has been
considered as a main cause of slope failure in many studies (Lenti and Martino, 2012; Gaudio
et al. 2014).
As also stated in sec. 3.3, hillslopes affected by the seismic shaking have also been prone to
rainfall induced failures, particularly in the form of debris flows. Further, the earthquake
induced shear strength degradation of slope material may also result in the enhanced
entrainment during a debris flow event (Liu et al. 2020). These debris flows might be initiated
either by increased pore pressure (or GW induced hydraulic pressure) or runoff involving
entrainment (Godt and Coe 2007). Though the GW effect is obtained on the slope instability
(Fig. 9, 10), potential response of the slope under excessive rainfall is explored through debris
flow runout analysis (Fig. 14, 15).
The debris flow runout predictions revealed a non-linear increase in the debris flow height
(9.0-26.0 m) and velocity (2.1-3.0 m/sec.) along the river channel on using the increasing
thickness (5,10, 15, and 20 m) of erodible material (Fig. 14). This non-linearity is attributed to
the downstream variation of the river channel width (Fig. 14c) and influx of debris flow
material from the slope. Though the present study noted the influence of channel morphology
on the debris flow characteristics, other studies have observed the changes in channel
morphology caused by the debris flows (Remaître et al., 2005; Simoni et al. 2020).
Thus, there seems to be a positive feedback process between channel morphology and debris
flow. This feedback notion is further strengthened by the finding of debris flow extent across
the river channel (Fig. 15d, e). At both of these locations, slope toe extends towards the E-SE
direction resulting in higher channel sinuosity. These extended slope toes probably represent
paleo-landslide and/or fluvial deposits. Signs of flow relics at the slope surface & failure at
slope toe at these locations (Fig. 2d,e) further support the possibility of paleo-landslide
deposit. Thus, the predicted extent of potential debris flow is found to follow the trails created
by previous landslide flow and/or slide events. Aforementioned findings, temporally
increasing rainfall, soil moisture, and surface runoff (sec. 2.2), and frequent debris flows/flash
floods in this region (Micu et al. 2013; Grecu et al. 2017; Micu et al. 2019) pose increasing
risk caused by debris flow in the study area.
Finally, there are still some uncertainties in such predictive approaches that are as follows; (1)
inclusion of subsurface discontinuity network, spatially varying groundwater surface, and
material heterogeneity in the 3D model, (2) inclusion of variable depth and phases in the
runout modeling. Despite these possible uncertainties, which will be overcome in future
prospects, such studies are required to minimize the risk and avert the possible disasters.
**6 CONCLUSIONS**
By utilizing field based data and numerical simulations of a massive (~9.1 Mm²) 'Varlaam'
landslide in the SE Carpathians (Romania), present study explored the potential response of
this landslide in seismic and rainfall regime.
The slope revealed the Factor of Safety (FS) in a range of 1.17-1.32 with a displacement of
0.4-4 m (under gravity load) that increases up to 8-60 m under seismic force. The
Groundwater (GW) further decreased the slope stability. The GW effect is attributed to the
hydraulic pressure in the joint against the overlying loose material that decreased the normal
stress and hence the shear strength of overlying loose material. Ground motion amplification,
during seismic shaking, further revealed the potential instability of slope with a Peak Ground
Acceleration (PGA) on the slope surface in a range of 0.65g - 1.88g. Such amplification
pertains to complex topography of landslide, soil thickness variation, and impedance contrast.
Further, though the GW effect is obtained on the slope instability, potential response of the
slope under excessive rainfall is also evaluated through debris flow runout analysis. The
predicted debris flow revealed a non-linear increase in the debris flow height (9.0-26.0 m) and
velocity (2.1-3.0 m/sec) along the river channel. This variation along the river channel is
attributed to the river channel morphology and influx of debris flow material from the slope.
Owing to the predictive nature of present study, the concept may be applied in other terrains
subjected to frequent landslides mostly triggered by extreme rainfall & earthquakes.
**Author contribution:** VK and HBH conceived the idea. All authors participated in the field
data collection & data interpretation. VK, LC, and ASM performed the numerical simulations.



MM led the geomorphic interpretation. All authors contributed to the writing of the final
draft.
**Competing interests:** The authors declare that they have no conflict of interest.
**Financial support**: Authors are thankful for the financial grant by the F.R.S.–FNRS Belgium
in the frame of the Belgian-Swiss collaboration project '4D seismic response and slope
failure'.
**ACKNOWLEDGEMENT**
Authors acknowledge Philippe Cerfontaine, Martin Depret, Nirmit Dhabaria and George
Catalin Simion for data acquisition (DGPS and seismological measurements).

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
