# Peer review of "A case study from the SE Carpathians, Romania"

_Natural Hazards and Earth System Sciences, 2021_

## Author Comment (AC1)

**Response to Referee #1**

1.      The description of the rainfall-related data is not sufficiently complete: more details should be provided in the text regarding how (instruments) and where (location with respect to the landslide site) rainfall, surface runoff and soil moisture were acquired.

**Response**: As per the suggestion, more details have been provided regarding these parameters in the revised MS. The location in which these parameters have been considered has been marked on Fig. 1 also for easy visualization.

2.      Data shown in Fig. 3 and Fig. 4 do not allow drawing conclusion of a clear increase in the trend of annual rainfall (and other related parameters) in the area that could favor landslide reactivations in the future. The increase mentioned in the text (line 120, Fig. 3 and Fig. 4) strongly depends on data binning: if one selects years 2001-2010 as the first bin for example and years 2011-2020 as the second one, the conclusions could be the opposite. More data analysis is therefore required: in particular, standard deviation values need to be added to the average values depicted in Fig. 4. Are the relative differences in annual rainfall (and other related parameters) averages (including their standard deviations) really meaningful between the two periods of times?

**Response**: In view of the reviewer' suggestion to incorporate the error values along with average values, we have updated Fig. 3 (attached herewith).

Though it is understandable that year specific binning, 2000-2009 and 2010-2019, might affect the pattern in a time period of years 2000-2019, year 2010 was related to 2010-2019 phase because it has been a year of paradigm shift in the recent climate. This shift can be understood from the fact that in year 2010, Northern Hemisphere surface temperature (combined global land and ocean) was the warmest on record, at 0.73°C above the 20th century average (https://www.ncdc.noaa.gov/sotc/global/201013, accessed on 19[th] Feb., 2021). This Northern Hemisphere temperature change has been related to the dramatic shift in the El Niño–Southern Oscillation, particularly to Negative-Arctic Oscillation (AO) during Dec.-Feb. in North America, Europe & Mediterranean region (Cohen et al. 2010; AMAP, 2011). Such Negative AO results in relatively higher precipitation in Southern Europe like in the SE Carpathians (study area) (AMAP, 2011).

Nonetheless, year specific binning has been avoided in the revised figure for better understanding. Further, the increase in rainfall & associated parametres that had been highlighted in the manuscript is justified using a longer time series (1982-2019). Fig. 3 and Fig. 4 have been merged for comprehensive understanding. As far as the notion of landslide reactivation is concerned, the present study proposes the possibility based on the exiting stability evaluations and recent events in the region (Micu et al. 2013; 2016; Micu, 2019).

[Figure]

**Fig. 3**: Rainfall (RF), Soil Moisture (SM), and Surface Runoff (RF) pattern. Dataset (except Fig. 3j) source: FLDAS_NOAH01_C_GL_M model (McNally et al. 2018). Spatial resolution of dataset: 0.1° (10 km). Data source of Fig. 3j: GPM IMERG Final Precipitation (Huffman et al. 2019). Spatial resolution of dataset: 0.1° (10 km). The rainfall data in Fig. 3j is available only from 1 June, 2000. The SM and SR data is not available at daily scale for the study area. Blue line (in Fig. a, d, g) indicates linear regression and shaded region around it refers to 95% confidence interval. Dots in box plots refer to outliers. Red line in Fig. 3j refers to extreme rainfall (30mm/24 h). Grey shaded region in Fig. 3k refer to those months that witnessed above average values of RF, SM, and SR.

**3.** Is the time scale of a year (line 122) appropriate to draw the conclusion of a link between rainfall and debris flows / flash floods given that « water related parameters » strongly vary over the year? How many debris flows and flash floods were registered during these two periods? Is it significantly larger than those observed over the other periods?

**Response**: We agree with the statement that "water related parameters strongly vary over the year" and such variation has been shown in the pattern of Soil Moisture and Surface Runoff also (Fig. 3f, i). However, in our statement, we related the year 2005 and 2010 debris flow/flash

floods with relatively higher values of rainfall & associated parameters based on observed real events (Micu et al. 2013; Grecu et al. 2017) and our own observation (Fig. 3b, h).

As far as the time scale is concerned, temporal initiation of debris flow may vary from few hours after rainfall to days depending upon the rainfall threshold and soil characteristics (Bacchini and Zannoni, 2003). Since the extreme rainfall in study area has been defined as >30mm/24 h (Apostol 2008; Croitoru et al. 2016) and there were 5 rainfall events having >30mm/24 h in both 2005 and 2010 years (Supp. Fig. R1), it is reasonable to relate year 2005 and 2010 debris flow/flash floods with relatively higher values of rainfall & associated parameters.

[Figure]

**Supp. Fig. R1:** Extreme rainfall (>30 mm/24 hr) vs. years (2000-2019).

Though year 2007 and 2016 also had 5 rainfall events having >30mm/24 h but these years didn't have as high surface runoff as year 2005 and 2010 (Fig. 3h). It is to note that many conceptual and physically based models have been proposed relating the initiation of debris flow to surface runoff conditions (Kean et al. 2013; Simoni et al. 2020). Further, the response of Soil Moisture is not so prominent because soils can store up to dozens or even hundreds of millimetres rainfall before reaching saturation (peak) and supporting surface runoff (Tramblay et al. 2010). The query regarding the number of debris flow events and their scale (or magnitude) in comparison to other events (if occurred) is difficult to respond at present because no field data exists for the study area in this context. We are hopeful to cover such aspects in future prospects.

4.      How comes that the relatively high amounts of rainfall in years 2013 and 2016 did not lead to increased surface runoff and soil moisture? The authors should provide additional information to support their conclusions.

**Response**: In view of reviewer' suggestion, the plot showing annual variation of rainfall has been revised with the inclusion of errors (Fig. 3b) and now annual trend is more inclusive to analyse owing to inclusion of monthly variation .

It is to note that the temporal coexistence of Rainfall, Surface Runoff, and Soil Moisture depend upon rainfall threshold and soil conditions (antecedent soil moisture). Though the year 2013 might have higher annual rainfall, it does have only two extreme rainfall (>30mm/24 h) events (Supp. Fig. R1) and the monthly variation in also relatively lower (Fig. 3b). Therefore, increased Surface Runoff, and Soil Moisture are not observed in this year.

The 2016, however, had 5 rainfall events having >30mm/24 h and resultant higher Surface Runoff, though not as high as in year 2005 and 2010 (Supp. Fig. R1, Fig. 3h). It is to note that Surface Runoff (water, from precipitation that flows over the land surface) correlates well with the Rainfall unlike the Soil Moisture that retains part of the rainfall before achieving saturation and hence do not correlate well, as noted in the study area (Supp. Fig. R2).

[Figure]

**Supplementary Fig. R2**: Correlation of Rainfall, Surface Runoff, and Soil Moisture at monthly scale during the years 1982-2019. Blue line indicates linear regression and shaded region around it refers to 95% confidence interval. Data Source: McNally et al. 2018.

The Soil Moisture didn't show any peak in year 2016 that can be understood from the fact that soils can store up to dozens or even hundreds of millimetres rainfall before reaching saturation (peak) and supporting surface runoff (Tramblay et al. 2010).  Further, Soil Moisture dynamics depend on various factors that vary spatio-temporally (Zhao et al. 2015) and detailed discussion might be beyond the scope of present study. We are hopeful to cover such aspects in future prospects.

**5.**      Why don't the authors comment on the distribution of earthquakes with respect to the location of the case study? How far the epicenters are from the landslide site? Can we expect them to have an impact on the triggering / reactivation of the landslide? Most earthquakes in the area have a magnitude smaller than 5. Besides these earthquakes are quite deep: can they trigger landslides?

**Response**: In view of the constructive query, we have updated Fig. 5 that also shows the distance of earthquake epicentres from the study area (Fig. 5d).

[Figure]

**Figure 5:** Earthquake pattern. (a-b) Position of study area (c) Depth and Earthquake magnitude (d) Distance of epicentres from landslide and Earthquake magnitude. Data source: National Institute for Earth Physics, Romania

It is appropriate to say that majority of the earthquakes are having magnitude smaller than 5 and quite deep (mostly between 60-180 km). However, most of these earthquakes have their epicentres within 50 km distance from the study area (Fig. 5d) that justifies the significance of considering dynamic evaluation of landslide slope. Such intermediate to deep earthquakes in Vrancea region (study area) have triggered landslides as far as 250–300 km from their epicentres (Havenith et al. 2016). Further, as we have emphasized in "Introduction" section of this study, we are of understanding that landslide triggering/reactivation does not happen in singularity. Rainfall and earthquakes both contribute to the slope instability and such slopes can result into failure during a trigger caused by any of these. It is to note that along with earthquake epicentres within 50 km, the region is also subjected to rainfall from different pressure lows and front systems carrying saturated winds from Mediterranean Sea, Black Sea and Atlantic Ocean (Mihailovici et al. 2006; ICPDR/IKSD 2012; Micu et al. 2013; Grecu et al. 2016).

**6.**     The authors intend to study the effects of rainfall and earthquakes on landslide occurrence in the study site: are there any established correlations between landslides and earthquakes or rainfall in this area? Have they changed over time? Further references could be added to illustrate the Varlaam landslide reactivations over time and their causes.

**Response**: We understand the reviewer' perspective for such query. The present study is intended to evaluate the hillslope response under extreme Rainfall and Earthquake conditions. More emphasis was given to understand the particular hillslope 'Varlaam' response. Further, correlating rainfall and earthquake at slope level might be difficult because both triggering factors operate and affect at different scale. Earthquakes even with the distant epicentres (250–300 km) have triggered landslides in the region (Havenith et al. 2016), whereas rainfall varies within kilometers (Supp. Fig. R3). Another limitation exists with the availability of high resolution rainfall data. At present, the best possible spatial resolution for the daily rainfall data, available for the study area, is 0.1 (10 km) (Huffman et al. 2019).

[Figure]

**Supplementary Fig. R3**: Rainfall and Earthquake distribution around study area. Daily Rainfall data is time averaged (2000-2019) to show spatial distribution. Rainfall Data Source: GPM IMERG Final Precipitation (Huffman et al. 2019). Earthquake Data Source: National Institute for Earth Physics, Romania.

Unfortunately, no such study exploring relationship of earthquake, rainfall, and landslides exists in the region at present. We are hopeful to cover such relationship at regional scale in future prospects.

**7.** The methodology needs to be outlined clearly, in more details. In particular, a better balance needs to be found between the description of the methodology and the results of both types of numerical approaches: landslide triggering modeling on the one hand with UDEC software and debris flow runout simulation on the other hand with RAMMS software. Additional data processing should be provided to support the interpretations and the conclusions

**Response**: As per the suggestion, methodology has been reconfigured in the revised MS.

**8.** The meaning of « potential material displacement » (line 204) needs to be clarified in the text. Do those displacements refer to : 1) Displacements after the model has reached static equilibrium (under gravity only) or 2) displacements obtained at the end of the factor of safety calculation using the shear strength reduction technique ? In both cases, the displacements are not meaningful in terms of absolute values.

**Response**: The potential material displacement under static condition refers to displacements after the model has reached static equilibrium under gravity load. The same explanation has been added in the revised MS. We are of understanding that this parameter and its values certainly have meaning. If 'absolute' term refers to mathematical meaning, we would like to clarify that we have presented and discussed displacement in a range and even errors are presented in Fig. 10: Comparison of material displacement under different conditions (original MS). If 'absolute' refers to singularity in general meaning, this is not the only parameter used to define slope stability. It is discussed along with Factor of Safety. This information will also be mentioned in the revised MS for better understanding.

**9.** Boundary conditions selected along the base of the model for the static and the dynamic runs should be better explained (line 200, Fig. 7). Why do the authors select x-viscous boundary for the dynamic simulations (and not x- and y-viscous boundaries as it is more often done?

**Response**: As per the suggestion, more information has been provided on boundary conditions in the revised MS. Further, we would like to clarify the boundary conditions used in the models. The lateral boundaries in all four slope sections (or models) were considered as free-field owing to near surface position of hillslope. The purpose of free-field boundary is to create a medium identical to infinite model so that reflection of seismic waves can be avoided in limited space (UDEC v.6, 2014). A stress-boundary condition (Joyner and Chen, 1975; UDEC v.6, 2014) was applied at the base in which horizontal direction is considered as viscous, whereas vertical direction is kept free. This stress-boundary condition converts seismic input from velocity wave to stress wave. Note that, though the seismic input is presented as Acceleration vs. Time in Fig. 8 in original manuscript, it was applied as velocity wave. As per the suggestion of reviewer, these details have been mentioned in the revised MS.

**10.** How representative are the seismic inputs (Fig. 8) of expected earthquakes in the region?

**Response**: This study involved following two seismic inputs; Ricker wavelet (Ricker 1943) and signal record of the 1976 Friuli Earthquake. As also mentioned in the MS, the Ricker wavelet, a theoretical waveform, provides an advantage to be a relatively short signal marked by an energy distributed over a range of frequencies. Therefore, the PGA and spectral ratio are evaluated using the Ricker wavelet to understand the ground motion amplification on the landslide surface. As far as signal representation for expected earthquake is concerned, it is difficult to comprehend owing to scarcity of reliable signals of big earthquakes in the Vrancea region. Thought the 1976 Friuli Earthquake signal might not be the most representative one, but the level of peak acceleration, of about 0.15g is representative for the Vrancea region.

**11.** How was groundwater introduced into the models? Did the authors conduct a fluid-flow calculation (either coupled or uncoupled with the mechanical stress calculation) prior to the factor of safety calculations to derive static initial stresses under wet conditions? How was water considered during the dynamic simulations? Did they use a static water table (no fluid flow) enhancing fluid pressure on the overlying medium and therefore reducing its factor of safety? If so, this should be clarified in the text.

**Response**: As rightly pointed out by reviewer, coupled hydraulic (fluid flow)-mechanical analysis was used in which mechanical deformation and joint fluid pressure affect each other as analysis progresses. The UDEC v.6 uses parallel-plate model (Witherspoon et al. 1980) to achieve such framework. Further, model was brought to static equilibrium before performing Factor of Safety calculations. It is to note that, FS calculations were also performed under mechanical stress only (without fluid flow). Steady-state (water table) fluid flow analysis was used to simulate fluid flow. As suggested, it has been described properly in the revised MS.

**12.** Is « partial loss of shear strength during seismicity » (line 449) explicitly taken into account in the modeling? If so, this should be described in details in the methodology.

**Response**: No, the statement is used as an inference based on our findings and previous studies supporting the possible cause (Novak and Yan, 1990; Safak, 2001).

**13.** The description of the RAMMS software in terms of governing equations could be enhanced. What kind of forces may act within the moving mass during the landslide propagation process?

**Response**: As per the suggestion, supplementary data involving governing equations (during runout, deposition) has been included in the revised MS.

**14.** How cohesive or non-cohesive is the expected flow? How can the modeling parameters be related to soil physical properties or to saturation conditions?

**Response**: The cohesion value of 0.001 MPa was used as an input for the release area. It is difficult to comprehend the nature of expected flow in terms of cohesion unless an analysis exploring the interrelationship of cohesiveness (clay content), dry coulomb friction ($\mu$), and turbulence ($\xi$) is performed. Owing to the possible vastness of this issue and scope of present study, we could not perform such analysis. Though cohesion might play an important role as the flow decelerates, especially at the deposition zone, cohesion is noted to vary between the flow front and tail. Further, owing to varying saturation & entrainment during the flow, frictional parametres might also vary and hence it has been difficult to relate the input parametres with the soil physical properties or to saturation conditions (Platezer et al. 2007; Zimmermann et al. 2020). Nonetheless, we are hopeful to cover such aspects in further studies.

**15.** A sensitivity analysis on the impact of the selected flow resistance parameters (friction and turbulent coefficient parameters) on the outputs (predicted runout zones and characteristics in terms of flow velocities and heights) could be added as other papers (see Zimmermann et al., 2020 - doi:10.3390/geosciences10020070 - for example) showed that results strongly depend on those parameters.

**Response**: As per the suggestion, sensitivity analysis has been performed to evaluate the possible influence of friction and turbulent coefficient parameters on the flow height, velocity, and runout extent. Further, in view of the main focus of this study, the result of this sensitivity analysis is presented & discussed as a supplementary figure in the revised MS.

**16.** What is the user selected stopping criteria for the debris flow simulation? What is its impact on the modeled deposition patterns?

**Response**: The stopping criterion was based on momentum threshold, which was considered as 5 % of moving mass. We are aware of the fact that values of this threshold along with that of frictional & turbulence parameters will surely affect the deposition pattern in terms of extent, flow height, pressure & velocity. Since performing a sensitivity analysis in this context could go beyond the scope of present study, it was chosen as 5% based on the default suggestion (RAMMS v.1.7.0, https://ramms.slf.ch/ramms/downloads/RAMMS_DBF_Manual.pdf). Further, this is one of the potential issues for future research prospects, particularly for depth-averaged, single phase simulations. We are hopeful to cover such aspects in further studies.

**17.** The understanding of the initial conditions (ie. quantity of expected entrainment material) leading to the debris flow occurrence is not straightforward: is it a block release that initiates the propagation?

**Response**: The initial condition (or release area) was used as an unchanneled flow (or block release). Though the landslide surface has some relics of flow channels near left flank (shown in Fig. 2d, e in manuscript), the data pertaining to the spatial-temporal pattern of discharge at these flow channel/gullies was not available. Therefore, the release area concept was chosen because it

has been more appropriate when the flow path (e.g. gully) and its possible discharge on the slope is uncertain (RAMMS v.1.7.0). The similar information is added in the supplementary data, in revised MS, involving all the details of runout simulation, performed in this study.

**18.** Do the values of « 5, 10, 15 or 20m of material depth » mean that from Fig. 7, all debris from top to the bottom of the landslide, down to a depth of 5, 10, 15 or 20m correspond to the initial release material for landslide runout simulations? If so, could the authors further quantify each event in terms of initial volume of debris in Fig. 14 d, g, j and m ? With this assumption of an initial erodible depth of loose materials, does this mean that the initial volume at release may not be continuous within the landslide (in particular for the 5m depth, Fig. 14 d to f) ? More details could be provided in the caption of Fig. 14b on the different depths used in the modeling.

**Response**: We would like to clarify that following depth ranges; 5, 10, 15 or 20m correspond to 60-80 m soil thickness region only, which can be seen in Fig. 14a, b in original MS. These different depths were considered as block release in view of uncertainties to ascertain the exact depth of loose material that will be eroded/entrained during the debris flow. As suggested, the figure caption has been improved for better understanding in the revised MS. We will also remove white space between values in the legends of this figure to avoid confusion.

**19.** Numerical modeling of the factor of safety and displacements considering various settings - In this part of the modeling, some paragraphs could be shortened - for instance, the impact of the internal friction angle on the factor of safety is well known ; same for the impact of a wet material vs a dry material – whereas others could be added - to strengthen the conclusions on ground-motion amplifications and shed light on the causes of such amplifications, purely topographic ground motion amplifications could be added along with 2D/1D aggravation factors.

**Response**: We appreciate the constructive suggestion. We would like to state that though these are well established facts in other case studies, our intent in the present study was to evaluate the response of a particular slope. FS and material displacement in dry & wet conditions constitute our objective along with topographic ground motion amplifications. Nonetheless, we will update the suggested sections in the revised MS.

**20.** In the outputs of the modeling of the factor of safety and displacements, the authors could comment more on the landslides locations/characteristics as a function of external loadings (either by rainfall only, earthquake only or both types of events, Fig. 9). Besides, using results from all the four 2D longitudinal cross sections, the authors could provide the readers with a first approximation on the overall 3D behavior of the landslide.

**Response**: We envisage that by using "landslides locations/characteristics", reviewer is suggesting discussing results with respect to position on the hillslope. If yes, we have already discussed our results using four different 2D slope sections (CS-1, CS-2, CS-3, and CS-4). CS-1 and CS-4 are positioned near the left and right flank, respectively, whereas CS-2 & CS-3 in the

middle. Nonetheless, we will add a paragraph in sec. 4.1.1 incorporating reviewer's suggestion regarding discussing results in view of loading factors in the revised MS. It is to note that though the rainfall effect is discussed using two different analyses; slope stability & runout simulation.

**21.** Numerical simulation of debris flow runout - Fig. 14f : could the authors comment on the very large peak of flow velocity on the left bank of the river ?

**Response**: The relatively high value of flow velocity in the upstream part of channel in Fig. 14f (in original MS) at 5m debris thickness can be understood using "turbulence (or Chezy resistance): ξ" factor used in the Runout governing equation (Voellmy 1955; Salm, 1993), mentioned below;

$$S_{f} = \mu\sigma + (\rho g U^2)/\xi \qquad\qquad \textbf{Eq. 1}$$

Details of these parametres can be seen in the governing equations of the RAMMS Debris Flow in the original MS. The Chezy resistance is famous as "turbulent" friction (Voellmy, 1955) in Debris Flow/Avalanche simulations since the mathematical formulations are similar to the well-known turbulent Chezy equation (Herschel, 1897; Chow, 1959) i.e., V=C √RS. Here, V is flow velocity, C is Chezy flow resistance coefficient, R is Hydraulic radius, and S is Slope gradient. Further, R= A/i where, A is Flow cross sectional area and i is Flow thickness. Therefore, in order to explain only the flow velocity, we utilized Chezy equation and shown that at low flow thickness (or debris thickness), flow velocity (debris flow velocity) are generally higher (Supp.Fig. R4c), as noted in Fig. 14f (in Original MS). Increase in flow thickness decrease hydraulic radius (Supp. Fig. R4a). Since velocity and hydraulic radius follow proportionality (Supp. Fig. R4b), lower flow thickness results in higher velocity (Supp. Fig. R4c). It is to note that values shown in Supplementary Fig. R4 are for pattern explanation purpose only. Slope angle and flow cross sectional area are kept constant.

[Figure]

**Supplementary Fig. R4**: Correlation of Flow thickness, Hydraulic radius, and Flow velocity.

**22.**   Why do the colorbars include blank color between for example different shades of blue?

**Response**: The blank spaces between colorbars have been removed to avoid any confusion in the revised MS, as suggested.

**23.**   How important is the hillslope topography and valley shape on the spreading of the debris flow?

**Response**: This query about 'how important' can be understood from the notion of 'Why is it important'. The influence of hillslope topography on the debris flow characteristics can be explained from the Eq. 1 i.e.,

$$S_{f} = \mu\sigma + (\rho g U^{2})/\xi$$

Here, $\sigma = \rho hg\cos(\psi)$. $\sigma$ is the normal stress on the running surface, $\rho$= density, g= gravitational acceleration, $\psi$ = slope angle, h= flow height. So Eq. 1 can be written as;

$$S_{f} = \mu\, \rho hg\cos(\psi) + (\rho g U^{2})/\xi$$

Now, steeper slopes (higher $\psi$ ) will decrease the normal stress and hence resultant shear resistance to flow ($S_{f}$). Decreased $S_{f}$ will result in accelerated debris flow on hillslope as per the principles of conservation of momentum (Voellmy 1955; Salm, 1993; Christen et al. 2010). The influence of "channel" shape on the debris flow can be seen in Fig. 14 (in original MS) where narrow sections possesses higher debris flow height, whereas wider sections will accommodate more debris flow and hence debris height will decrease. However, the relationship is not linear in our case because the debris flow along the channel will get influx from the hillslope flow that in turn will be subjected to hillslope topography. For more details, a supplementary figure showing the relationship of channel width and debris flow characteristics has been added in the revised MS.

**24.**   The authors mention that « runout findings … follow the same spatial extent as possibly followed by previous landslide events »: could they add information / plots to support their conclusions.

**Response**: We would like to state that the particular statement is an inference based on the predicted runout extent (Fig. 15d, e in original MS) and existing landform. The sinuosity of river at two locations, where predicted runout extended beyond the river channel, is related to the deposition possibly formed by previous debris flow events. This inference also considered flow relics/gullies near left flank implying the previous hillslope debris flow events. More detailed understanding can only be achieved by performing dating of sediments of the deposition and flow relics. However, possibility of multiple phase of flow and source-deposition delineation may limit the reliability of event dating. Such research problem may be considered in future research prospects. As suggested this information has been added in the relevant section.

**25.** The paper ends with a summary that provides an overview of the main results. It could be shortened to leave more space for a discussion (that is currently reduced to lines 633-637).

**Response**: As per the suggestion, summary is shortened in the revised MS.

**26.** In the abstract, the modeling approaches used in this paper should be mentioned.

**Response**: As per the suggestion, 'Abstract' is updated in the revised MS.

**27.** Line 36: Froude and Petley (2018) studied « fatal non-seismic landslides ». Please select another reference to show the need for such a study.

**Response**: As per the suggestion, relevant citing is updated in the revised MS.

**28.** Fig. 3: What is the purpose of a polar plot representation to illustrate time evolution of annual rainfall? Besides, it is counter intuitive that past is in the right and present in the left. Average monthly rainfall could be added in each plot of Fig. 3b to ease the link with

**Response**: This figure has been updated (attached in this response document) in view of reviewer's comment no. 2 that required more details regarding these datasets.

**29.** Fig. 4. Meaning of Spatial resolution: 0.1° in the caption?

**Response**: It refers to 10 km. This information is added in the relevant caption.

**30.** Fig. 4f: Please add the time axis to ease reading of the plot

**Response**: This figure has been updated as Fig. 3 (attached in this response document) in view of reviewer's comment no. 2 that required more details regarding these datasets.

**31.** Line 392: This effect is attributed to the shear strength reduction approach » : this statement is not correct. The increase of the factor of safety is a consequence of an increase in the shear strength of the soil as a consequence of the increase in the angle of internal friction of the soil.

**Response**: We would like to clarify that "Shear Strength Reduction (SSR)" approach that is used to determine the Factor of Safety implies the similar meaning what the reviewer is stating. The line refers to increase in the FS owing to increase in Angle of internal friction. The SSR approach states that the Factor of Safety (or Strength Reduction Factor) is a ratio of Existing Shear Strength and Shear Strength at failure (Matsui and San, 1992; Griffiths and Lane, 1999).

**32.**    Fig. 12: Because plots are hardly readable, respective values of PGA could be added on top of each subplot.

**Response**: As per the suggestion, relevant figure (Acceleration vs. Time) is updated in revised MS.

**33.**    Paragraph 4.1.4: why not showing a transfer function plot that would be more informative than several curves of spectral ratios at given locations along the slope surface?

**Response**: To understand the response of the medium to the input signal, we compared the signals obtained in the monitoring points at the surface with the signal at the monitoring point at depth (base). This comparison is held in the frequency domain by computing the spectral ratios H(f) between the signal spectra at the surface stations and the signal spectrum at the depth monitoring point.

$$H(f)= S\_i(f)/ S\_base (f)$$

Where S_i(f) is the fourier transform at the ith monitoring point at the surface and S_base(f) is the fourier transform of the signal at the monitoring point at depth. Following McCowan and Lacoss, (1978), the "spectral ratios" computed in paragraph 4.1. and shown in figure therefore represent the transfer functions. We agree that this point was not clear in the manuscript and we will rephrase it accordingly in the revised MS.

**34.**    Fig. 5b : this zoom plot does not provide additional information with respect to plot 5a

**Response**: As per the suggestion, relevant figure is updated in revised MS.

**35.**    Fig. 7 : the possible locations of the water table inside the model are not clear to this reviewer

**Response**: As also explained in the response of comment no. 11, the UDEC allows the GW simulation through the joints only as per the parallel plate model. The depictions in the models (CS-1, 2, 3, 4) through the debris are for simplification purpose only. In increase of the GW level (as shown in Fig. 7d in the original MS) represents the increase in the pressure difference in the joint as per Cubic law or Parallel-Plate model (Witherspoon et al. 1980). Nonetheless, it is zoomed further in the revised MS for better understanding.

**36.** Fig. 8: plots showing the ranges of frequencies of the selected input motions could be useful.

**Response**: As per the suggestion, following figure is added in revised MS.

[Figure]

**Fig. 8**: Seismic signals in time & frequency domain. (a &b) Ricker Wavelet (as recorded at the model base monitoring point) (c & d) 1976 Friuli Earthquake, (Italy). Note: Different time scale.

**37.** Fig. 10d: can the authors comment on plot 10c where mean displacements in dynamic-dry conditions are larger than those in dynamic-wet conditions? The authors could comment on the respective role of PGA and the frequency contents of the input motions on the final displacements. To support this point, showing the frequency contents of the inputs could help.

**Response**: This query refers to the dry & wet dynamic (Friuli) conditions (highlighted in red rectangle in the figure below). We would like to clarify that though the 'median' displacement during dry-dynamic stage is relatively more than the wet-dynamic, maximum displacement in wet-dynamic condition is still higher. This lowering of median in wet-dynamic condition is caused by longer range of the displacement that is more comprehensible in Fig. 9o, 9s (in original MS). During the wet-dynamic

condition, range of displacement spreads further down the debris.

**38.** Fig. 11: the extreme values of the vertical axis are not appropriate in plots a to d (and in plots e to h). Plots from i to l do not allow for an easy quantification of the increase or decrease of displacements as a function of slope parameters (for instance elastic modulus) or conditions. In plot d, the elastic modulus refers to which parameter of the soil?

**Response**: In view of the suggestion, vertical axis in Fig. 11 is updated for better representation of pattern in the revised MS. Plot 'd' refers to 'Young' modulus (or elasticity modulus'. As mentioned in the MS, the Young's modulus (or elasticity modulus) along with other parametres used in the parametric analysis is one of the main controlling parametres for Factor of safety and material displacement relatively.

**39.** Fig. 12 : please add the horizontal axis (time) on all subplots

**Response**: As per the suggestion, it is added in the revised MS.

**40.** Fig. 13: which displacements are reported: surface or inside the landslide mass displacements?

**Response**: Fig. 13 presents spectral ratio pattern. If the question pertains to Fig. 9, 10, displacement refers to landslide mass. Since the UDEC modeling allows the internal deformation of blocks through zoning, we have utilized that to represent the displacement in landslide mass.

**41.** Fig. 15: the quality of the insets plots could be improved.

**Response**: As per the suggestion, it is improved in the revised MS.

**42.** Table 1 : unit for shear stiffness could be added (in addition to kn/10)

**Response**: As per the suggestion, it is added in the revised MS.

**Response to Referee #2**

This study investigates a pertinent research question i.e. assessment of a combined trigger for landslides. Evaluation of stability at the source zone and prediction of run out in case of failure is quite interesting aspect of this study. The stability has been assessed both for static and dynamic conditions. The article is written well. I have following minor comments:

**Response**: We are encouraged by the reviewers' constructive assessment and comments. Each suggestion/comment is addressed below with full consideration.

**1.**     Line 67: Do you have any idea when this landslide happened? A historical perspective of the landslide will be helpful to the readers.

**Response**: The earliest record of this landslide is mentioned in the geological map by Murgeanu et al. (1965). Unfortunately, no previous record and/or dating are available at present. This historical constraint has been added in the relevant section for information of readers.

**2.**     Figure 1: Follow symbology for Anticline/Syncline as per the structural geology norms.

**Response**: As per the suggestion, symbols have been updated in the revised manuscript.

**3.**     Figure 4: Can you explain why all three peaks are only in 2005 and 2010, not for other years?

**Response**: We would like to answer this query in two phases in view of the source of variables (rainfall, soil moisture, and runoff). The first one explains the cause of rainfall peaks in year 2005 and 2010, whereas second one will seek the cause of peaks in surface runoff and soil moisture.

The year 2005 and 2010 had abnormally high precipitation due to synoptic conditions that involved pressure lows and front systems moving along a SE–NW trajectory from the Mediterranean Sea and Black Sea towards Central Europe and in west to east direction from the Atlantic Ocean to Eastern Europe. These trajectories led to severe flood and slope failure events in different parts of the Central and Eastern Europe (Mihailovici et al. 2006; ICPDR/IKSD 2012; Micu et al. 2013; Grecu et al. 2016). The influence of these trajectories is also visible in the regional rainfall pattern (Supp. Fig. R5), where year 2005 and 2010 have relatively higher rainfall in the study area in comparison to other years.

[Figure]

**Supplementary Fig. R5**: Regional rainfall variation. Inset 'a' shows the location of study area and extent used to represent elevation and regional rainfall variation. Image Source: Google earth. Inset 'b' shows regional elevation map. Data Source: SRTM. Rainfall data source: GPM_3IMERGDF v.06 (Huffman et al. 2019).

Now, the peaks of surface runoff and soil moisture owe their nature to the extreme rainfall in these years because the surface runoff and soil moisture data are based on the FLDAS (Famine Early Warning Systems Network Land Data Assimilation System) model (McNally et al. 2018). It utilizes precipitation datasets & analyses like CHIRPS (Climate Hazards Group InfraRed Precipitation with Station data) & MERRA2 (Modern-Era Retrospective analysis for Research and Applications, Version 2) along with land cover data to derive variables like soil moisture, runoff, and streamflow.

Generally, surface runoff (water, from precipitation that flows over the land surface) correlates well with the rainfall unlike the soil moisture that retains part of the rainfall before achieving saturation and hence do not correlate well, as noted in the study area (Supp. Fig. R2). Soils can store up to dozens or even hundreds of millimetres rainfall before reaching saturation, depending on porosity, depth and initial soil moisture (Tramblay et al. 2010). However, during extreme rainfall events, soil gets saturated and along with surface runoff, soil moisture reflects peak in its trend, as noted in our study during the years 2005 and 2010.

[Figure]

**Supplementary Fig. R2**: Correlation of Rainfall, Surface Runoff, and Soil Moisture at monthly scale during the years 1982-2019. Blue line indicates linear regression and shaded region around it refers to 95% confidence interval. Data Source: McNally et al. 2018.

**4.**     Figure 5: This is a good figure since it demonstrates the influence of earthquake as a triggering factor. Similar to this spatial representation of earthquake epicenter, you can consider showing the spatial distribution of rainfall pattern over the region.

**Response**: In view of the reviewer's insightful suggestion, we have prepared another figure (Supp. Fig. R5) highlighting the spatial distribution of rainfall over the region for the years 2000-2019.

**5.**     Line 145: Pl. indicates how many earthquakes occurred in the rainy season. This is important since the research question hinges around this statement.

**Response**: As per the suggestion, we have added another figure (Supp. Fig. R6) highlighting the monthly distribution of earthquake events for the years 1960-2019.

[Figure]

**Supplementary Fig. R6**: Earthquake ($4<M_w<8$) events during the years 1960-2019. 'a' shows monthly rainfall distribution during the years 1982-2019. 'b' & 'c' represent earthquakes in different months during the years 1960-2019. Shaded region on the top & right of the section 'c' highlight density of the data on respective axis. Data Source: National Institute for Earth Physics, Romania.

**6.** Line 170: Is it called as resonance frequency or peak/predominant frequency in Nakamura analysis?

**Response**: We agree with the fact that Nakamure (1991) has used the word "predominant" for the frequency ($f_o$) in eq. 1 of the manuscript i.e., ($h=V_s/4f_o$). However, this predominance owes its nature mostly due to the resonance effect and hence it has also been termed as 'resonance frequency' (Murphy et al. 1971; Beresnev and Wen, 1996).

**7.** Line 178: Mention how the shear wave velocity was estimated? What type of investigation was done: CPT? Some details are required here.

**Response**: We would like to clarify that the shear-wave velocity ($V_s$) values in the present study are based on Mreyen et al. (2021), in view of the similar litho-tectonic conditions and spatial proximity. For the loose overburden (soil) and rockmass, the $V_s$ are taken as ~400 m/sec and ~900 m/sec, respectively

**8.**     Line 183: High resolution (5 m) TanDEM-X DEM data are available. Why that has not been used?

**Response**: We understand the reviewer' perspective regarding possible usage of the TanFDEM (~6 m) instead of TanDEM-X (~12 m) that we used. We are of understanding that the pixel size of TanDEM-X is reasonable in view of the size of such a huge (9.1 Mm$^2$) landslide and finer resolution, as suggested, could have made the simulations more complex. Though it is true that better spatial resolution of the 'TanFDEM' could be helpful, it generally comprises relatively more random height error, particularly for slopes $\geq$ 20° (Wessel, 2018). Random error is generally caused by thermal noise and residual geometric decorrelation effects. Geometric decorrelation (also known as baseline decorrelation) is a limiting factor in the performance of interferometers that feature large baselines. We are not sure at this stage about the possible influence of such random height error on the topography that in turn could have affected seismic signals. However, we are hopeful to utilize the suggested DEM and perform a comparative evaluation in future prospects.

**9.**     Figure 6: The 80 m soil thickness is quite huge. Since it is mathematically estimated, better to so some validation result also.

**Response**: We would like to clarify that this thickness was ascertained based on field based array method (sec. 3.1 of original MS). Recently, Mreyen et al. (2021) have also inferred the debris/soil thickness of a landslide in this region (within 10-15 km) in the range of 70-90 m based on geophysical methods.

**10.**    Line 270: Better to write 'cross sections'

**Response**: As per the suggestion, terminology has been updated in the revised manuscript.

**11.**    Line 294: Check the symbol for slope angle.

**Response**: As per the suggestion, symbol has been updated in the revised manuscript.

**12.**     Line 520: Please explain why you selected release depth for run out starting from 5 m contrary to large overburden depth present in this area. Also explain how did you identify the release area?

**Response**: We would like to clarify that identification of the release area was based field and satellite imagery observations. Following four factors; gullies (Fig. 2c), flow relics (Fig. 2d), signs of failure (Fig. 2e), and overburden thickness pattern (Fig. 6c) were considered while selecting the release area. The thickness region of 60-80 m having flow relics and signs of failure were therefore selected as the potential release area (Fig. 14b). We understand that there could be many possibilities of major/minor release area, apart from the region we selected, in the landslide surface depending upon the pore pressure, entrainment rate during the debris flow triggering. However, the best possible region was chosen in view of field and satellite imagery observations.

As far as thickness in concerned, it is reasonable to consider that during the slope failure, irrespective of type of trigger, entire loose material might not slide down, and hence depth was taken as a variable ranging from 5 to 20 m. Again there could be many depth ranges between 5 and 80 m depending on the objective but performing simulations on all such ranges might have changed the focus of present study. Though the present study eliminated the uncertainties caused by a particular depth value, scope for further research in this context is still open for future prospects. Further, it has been noted in the field observations that such events in this region generally initiate with shallow thickness of overburden.

**13.**     Line 523-535: Present this paragraph as a Table.

**Response**: As per the suggestion, relevant paragraph has been presented as a table in the revised manuscript.

**14.**     The range of factor of safety is >1, which indicates stability of the landslide in general. In this context, explain what could be the extreme trigger condition to reduce the factor of safety below 1

**Response:** The Varlaam landslide hillslope, despite being an active landslide, attained the FS >1. Following reasons justify this nature of the FS;

(1) The FS has been defined as a ratio of the existing shear strength and shear strength at failure under the static condition (Matsui and San, 1992; Griffiths and Lane, 1999). The existing shear strength, particularly the angle of internal friction of slope material may decrease during the extreme rainfall induced percolation and the earthquake induced particle movements (Cai and Ugai, 2004; Chang and Taboada, 2009). As a result, the FS may decrease further implying a growing instability;

(2) The FS is a response of the entire slope. It means that it integrates the response of all zones (FDM zones) in the slope, whereas in reality, the failure might also occur locally. Therefore, along with the FS, pattern of the displacement was also considered to infer the instability regime.

We are hopeful that the reviewers will understand the merits and limitations of the predictive nature of our approach that we tried to present judiciously

**References**

Apostol, L. (2018). The Mediterranean cyclones–the role in ensuring water resources and their potential of climatic risk, in the east of Romania. Present environment and sustainable development, 2, 143-163.

Arctic Monitoring and Assessment Programme (AMAP) (2011). Snow, Water, Ice and Permafrost in the Arctic (SWIPA), Climate Change and the Cryosphere, Oslo, Norway.

Bacchini, M., & Zannoni, A. (2003). Relations between rainfall and triggering of debris-flow: case study of Cancia (Dolomites, Northeastern Italy). Natural Hazards and Earth System Sciences, 3(1/2), 71-79.

Beresnev, I. A., & Wen, K. L. (1996). Nonlinear soil response—A reality?. Bulletin of the Seismological Society of America, 86(6), 1964-1978.

Cai, F., & Ugai, K. (2004). Numerical analysis of rainfall effects on slope stability. International Journal of Geomechanics, 4(2), 69-78.

Chang, K. J., & Taboada, A. (2009). Discrete element simulation of the Jiufengershan rock-and-soil avalanche triggered by the 1999 Chi-Chi earthquake, Taiwan. Journal of Geophysical Research: Earth Surface, 114(F3).

Chow, V.T. (1959). Open-channel hydraulics. McGraw-Hill civil engineering series.

Christen, M., Kowalski, J., & Bartelt, P. (2010). RAMMS: Numerical simulation of dense snow avalanches in three-dimensional terrain. Cold Regions Science and Technology, 63(1-2), 1-14.

Cohen, J., Foster, J., Barlow, M., Saito, K., & Jones, J. (2010). Winter 2009–2010: A case study of an extreme Arctic Oscillation event. Geophysical Research Letters, 37(17).

Croitoru, A.E., Piticar, A. and Burada, D.C. (2016). Changes in precipitation extremes in Romania. Quaternary International, 415, 325-335.

Grecu, F., Zaharia, L., Ioana-Toroimac, G. and Armaş, I. (2017). Floods and flash-floods related to river channel dynamics. In Landform dynamics and evolution in Romania (pp. 821-844). Springer, Cham.

Griffiths, D. V., & Lane, P. A. (1999). Slope stability analysis by finite elements. Geotechnique, 49(3), 387-403.

Havenith, H. B., Torgoev, A., Braun, A., Schlögel, R., & Micu, M. (2016). A new classification of earthquake-induced landslide event sizes based on seismotectonic, topographic, climatic and geologic factors. Geoenvironmental Disasters, 3(1), 1-24.

Herschel, C. (1897). On the origin of the Chezy formula, Journal Association of Engineering Societies, (18) 363-368.

Huffman, G.J., Stocker, E.F., Bolvin, D.T., Nelkin, E.J. and Jackson T. (2019). GPM IMERG Final Precipitation L3 1 day 0.1 degree x 0.1 degree V06, Edited by Andrey Savtchenko, Greenbelt, MD, Goddard Earth Sciences Data and Information Services Center (GES DISC), Accessed: Sep. 5, 2020, 10.5067/GPM/IMERGDF/DAY/06.

ICPDR/IKSD (2012). 2010 Floods in the Danube River Basin. Brief overview of key events and lessons learned International Commission for the Protection of the Danube River, Vienna

Joyner, W. B., & Chen, A. T. (1975). Calculation of nonlinear ground response in earthquakes. Bulletin of the Seismological Society of America, 65(5), 1315-1336.

Kean, J. W., McCoy, S. W., Tucker, G. E., Staley, D. M., & Coe, J. A. (2013). Runoff-generated debris flows: Observations and modeling of surge initiation, magnitude, and frequency. Journal of Geophysical Research: Earth Surface, 118(4), 2190-2207.

Kumar, V., Gupta, V., & Jamir, I. (2018). Hazard evaluation of progressive Pawari landslide zone, Satluj valley, Himachal Pradesh, India. Natural Hazards, 93(2), 1029-1047.

Matsui, T., & San, K. C. (1992). Finite element slope stability analysis by shear strength reduction technique. Soils and foundations, 32(1), 59-70.

McCowan, D. W., & Lacoss, R. T. (1978). Transfer functions for the seismic research observatory seismograph system. Bulletin of the Seismological Society of America, 68(2), 501-512.

McNally A. (2018). FLDAS Noah Land Surface Model L4 Global Monthly 0.1 x 0.1 degree (MERRA-2 and CHIRPS), Greenbelt, MD, USA, Goddard Earth Sciences Data and Information Services Center (GES DISC), 10.5067/5NHC22T9375G.

Micu, M. (2019). Landslide hazard assessment in Vrancea seismic region (Curvature Carpathians of Romania): achievements and perspectives. In First EAGE Workshop on Assessment of Landslide and Debris Flows Hazards in the Carpathians, 2019 (1), 1-5.

Micu, M., Bălteanu, D., Micu, D., Zarea, R. and Raluca, R. (2013). Landslides in the Romanian Curvature Carpathians in 2010. In Geomorphological impacts of extreme weather, Springer, Dordrecht, 251-264.

Micu, M., Jurchescu, M., Șandric, I., Mărgărint, C., Zenaida, C., Dana, M., Ciurean, R., Ilinca, V., Vasile, M. (2016). Natural Risks - Mass Movements, in M. Radoane, A.Vespremeanu-Stroe (Eds.) Landform dynamics and evolution in Romania, Springer, 765-820.

Mihailovici, M., Gabor, O., Rândașu, S., & Asman, I. (2005). floods in Romania. Hidrotehnica, 51(6), 23-35.

Mreyen, A. S., Cauchie, L., Micu, M., Onaca, A., & Havenith, H. B. (2021). Multiple geophysical investigations to characterize massive slope failure deposits: application to the Balta rockslide, Carpathians. Geophysical Journal International, 225(2), 1032-1047.

Murgeanu, G., Dumitrescu, I., Sandulescu, M., Bandrabur, T. and Sandulesu, J.: Harta geologică a RS România. L-35-XXI, scara 1: 200.000, Foaia Covasna, 1965.

Murphy, J. R., Davis, A. H., & Weaver, N. L. (1971). Amplification of seismic body waves by low-velocity surface layers. Bulletin of the Seismological Society of America, 61(1), 109-145.

Nakamura, Y. (2009). Basic structure of QTS (HVSR) and examples of applications. In Increasing seismic safety by combining engineering technologies and seismological data (pp. 33-51). Springer, Dordrecht.

Novak, M., & Han, Y. C. (1990). Impedances of soil layer with boundary zone. Journal of geotechnical engineering, 116(6), 1008-1014.

Platzer, K., Bartelt, P., & Kern, M. (2007). Measurements of dense snow avalanche basal shear to normal stress ratios (S/N). Geophysical Research Letters, 34(7).

Ricker, N. (1943). Further developments in the wavelet theory of seismogram structure. Bulletin of the Seismological Society of America, 33(3), 197-228.

Şafak, E. (2001). Local site effects and dynamic soil behavior. Soil Dynamics and Earthquake Engineering, 21(5), 453-458.

Salm, B. (1993). Flow, flow transition and runout distances of flowing avalanches. Annals of Glaciology, 18, 221-226.

Simoni, A., Bernard, M., Berti, M., Boreggio, M., Lanzoni, S., Stancanelli, L. M., & Gregoretti, C. (2020). Runoff-generated debris flows: Observation of initiation conditions and erosion–deposition dynamics along the channel at Cancia (eastern Italian Alps). Earth Surface Processes and Landforms, 45(14), 3556-3571.

Tramblay, Y., Bouvier, C., Martin, C., Didon-Lescot, J. F., Todorovik, D., & Domergue, J. M. (2010). Assessment of initial soil moisture conditions for event-based rainfall–runoff modelling. Journal of Hydrology, 387(3-4), 176-187.

Voellmy, A (1955). Über die Zerstörungskraft von Lawinen. Schweiz. Bauztg.

Wessel, B. (2018). TanDEM-X Ground Segment – DEM Products Specification Document", EOC, DLR, Oberpfaffenhofen, Germany, Public Document TD-GS-PS-0021, Issue 3.2, 2018. [Online]. Available: https://tandemx-science.dlr.de/

Witherspoon, P. A., Wang, J. S., Iwai, K., & Gale, J. E. (1980). Validity of cubic law for fluid flow in a deformable rock fracture. Water resources research, 16(6), 1016-1024.

Yadav OP (2000) An evaluation report on investigation, instrumentation and monitoring of landslides in Himachal Pradesh. Department of Science and Technology, New Delhi , India.

Zhao, N., Yu, F., Li, C., Zhang, L., Liu, J., Mu, W., & Wang, H. (2015). Soil moisture dynamics and effects on runoff generation at small hillslope scale. Journal of Hydrologic Engineering, 20(7), 05014024.

Zimmermann, F., McArdell, B. W., Rickli, C., & Scheidl, C. (2020). 2D runout modelling of hillslope debris flows, based on well-documented events in Switzerland. Geosciences, 10(2), 70.